# Fertilization Induced Soil Microbial Shifts Show Minor Effects on *Sapindus mukorossi* Yield

**DOI:** 10.3390/microorganisms13010173

**Published:** 2025-01-15

**Authors:** Juntao Liu, Zhexiu Yu, Yingyun Gong, Jie Chen, Ling Zhou, Weihua Zhang, Liming Jia

**Affiliations:** 1Key Laboratory of Silviculture and Conservation of the Ministry of Education, College of Forestry, Beijing Forestry University, Beijing 100083, China; ljt1120@bjfu.edu.cn (J.L.); yzhexiu@bjfu.edu.cn (Z.Y.); oxalis623@163.com (Y.G.); zhouling89757@bjfu.edu.cn (L.Z.); 2National Energy R&D Center for Non-Food Biomass, Beijing Forestry University, Beijing 100083, China; 3National Innovation Alliance of Sapindus Industry, Beijing Forestry University, Beijing 100083, China; 4Guangdong Provincial Key Laboratory of Silviculture, Protection and Utilization, Guangdong Academy of Forestry, Guangzhou 510520, China; 5Research Institute of Tropical Forestry, Chinese Academy of Forestry, Guangzhou 510520, China; chenjiecaf@hotmail.com

**Keywords:** bacterial community, diversity, fertilization, *Sapindus mukorossi*, soil fertility, structural distribution

## Abstract

Fertilization can improve soil nutrition and increase the yield of *Sapindus mukorossi*, but the response of soil microbial communities to fertilization treatments and their correlation with soil nutrition and *Sapindus mukorossi* yield are unclear. In order to investigate the characteristics of soil physicochemical qualities and the bacterial community, we carried out a field experiment comparing various quantities of nitrogen (N), phosphorus (P), and potassium (K) fertilizers to the unfertilized control treatments and the yield of *Sapindus mukorossi* in raw material forests in response to different applications of fertilizers and to try to clarify the interrelation among the three. Results showed that (1) there are significant differences in the effects of different fertilization treatments on the soil properties of *Sapindus mukorossi* raw material forests. The increase in the application rates of nitrogen or phosphorus fertilizers significantly reduced the soil pH value. (2) Compared with control, the α-diversity of bacterial communities was significantly lower in N_3_P_2_K_2_ and N_1_P_1_K_2_ treatments. Among the dominant groups of soil bacteria at the phylum level, the relative abundance of *Chloroflexi* showed an increase and then a decrease trend with the increase in N application. The relative abundance of *Firmicutes*, *Bacteroidota*, and *Fusobacteriota* was positively correlated with the application of P and K fertilizers, while the relative abundance of *Acidobacteriota* and *Verrucomicrobiota* decreased with the increase in P and K fertilizers. (3) The N_2_P_2_K_2_ treatment produced the highest *sapindus* yield (1464.58 kg/ha), which increased by 258.67% above the control. (4) Redundancy analysis (RDA) showed that the primary determinants of bacterial community structure were soil pH, total K, and effective P concentration. (5) Structural equation modeling (SEM) showed that soil nutrient content was the main direct factor driving the yield of *Sapindus mukorossi*, whereas the bacterial community attributes (e.g., diversity and structure) had minor effects on the yield. In summary, the rational use of formulated fertilization can change the bacterial community structure, improve the bacterial diversity, and increase the soil nutrient content, with the latter exerting a significant effect on the improvement of the yield of *Sapindus mukorossi*.

## 1. Introduction

A type of energy forest belonging to the genus *Sapindus* in the family Sapin-daceae, *Sapindus mukorossi* is primarily found in tropical and subtropical areas [1,2,3]. Due to the growing demand for energy worldwide in recent years, bioenergy species have gained a lot of interest as a new and promising energy source [4,5]. Many industries use *Sapindus mukorossi*, including biomedicine, bioenergy. and cosmetics. In traditional Chinese medicine, the entire plant of *Sapindus*—fruit, root, bark, and leaves—is utilized. It is a woody oilseed tree species that has been marketed for usage in recent years. Its soap pods contain roughly 22% crude protein, and the kernels of the soap berries contain roughly 40% fatty acids. The planting area is 20,000 ha in Fujian Province alone, which is one of its primary production locations [4]. However, the production and quality of bio-energy species are significantly influenced by the nutritional conditions of the soil. Massive nutrients like nitrogen (N), phosphate (P), and potassium (K) are essential for plant growth and are crucial for crop productivity and the synthesis of oil products. In order to achieve high yields and high quality of economic tree species, it is effective to apply N, P, and K fertilizers rationally [6,7]. However, farmers usually over-apply N, P, and K fertilizers in order to rapidly increase the yield of *Sapindus mukorossi* fruits [8,9]. Unexpected fertilization can result in major environmental issues, including groundwater source pollution as well as issues like sloughing, acidification, and deterioration of soil quality, all of which lower crop yields [10,11]. Exploring scientific fertilization techniques is essential to enhance the sustainable development of the *Sapindus mukorossi* industrial forest.

Fertilization has a direct impact on the physicochemical characteristics of soil [12] and it has been demonstrated that balanced NPK fertilization significantly influences the physicochemical characteristics of crops. increases the amount of organic matter and quick-acting nutrients in the soil, encourages root growth and inter-root soil microbial growth, and ultimately influences yield. [2,7,12,13]. Proper fertilization management can maximize crop yield [7]. However, overuse of fertilizer will result in soil acidification and severe soil nutrient imbalance, which will then create leaf nutrient imbalance and ultimately impact production and economic efficiency [11,14]. Continuous fertilization can increase soil organic carbon content, total nitrogen content, and soil productivity [15,16]. Therefore, fertilization not only affects soil properties but also influences microbial diversity in soil.

Soil microorganisms have a strong connection with soil physicochemical properties and are drivers of processes such as organic matter decomposition, nutrient cycling, and humus formation [17]. Since soil microorganisms are highly sensitive to environmental changes [18,19], a significant amount of research has been conducted on the impacts of nutrient addition or deficiency on soil microbial communities [20,21]. It was demonstrated that N application was able to significantly alter the bacterial community, soil properties, and yield of sugarcane (*Saccharum officinarum*). In particular, bacterial species richness and evenness were higher in the moderately fertilized treatment than in the control without fertilization [22,23]. Ma et al. [23] found that nitrogen fertilization increased the number of nitrifying bacteria but did not affect the number of other flora. Phosphorus fertilizer application decreased the number of nitrate-reducing bacteria and archaeal ammonia-oxidizing bacteria and increased the number of nitrifying bacteria and nitrous oxide-reducing bacteria. Tang et al. [24] showed that N and P additions altered the abundance and community structure of nitrification and denitrification genes in a subtropical Chinese fir plantation forest, and that the combined application of nitrogen and phosphorus drove changes in the ammonia-oxidizing archaeal community. Li et al. [7] reported a significant decrease in soil bacterial diversity after N addition, while potassium fertilizer had relatively little effect on soil bacteria. Conversely, N deficiency will limit the increase in soil microbial biomass and reduce community diversity [25]. It has been demonstrated that combined application of N, P, and K elevated *Panax ginseng* (*Panax ginseng C. A. Mey.*) yield and saponin content by increasing the relative abundance of potentially beneficial fungi and decreasing the relative abundance of potentially pathogenic fungi [6]. Therefore, NPK fertilizer addition can drive changes in soil bacterial community structure, abundance, and function by altering soil physicochemical properties. Overall changes in soil physicochemical properties and microbial community structure and diversity were observed after fertilizer application, but how these changes relate to fruit yield of plants still needs to be further investigated.

Presently, the area of plantation of *Sapindus mukorossi* in China is vast, and the economic benefits are considerable [3,26]. Fertilization is the main measure to regulate the yield and quality of *Sapindus mukorossi;* due to that, the soil fertility can be improved and soil productivity can be restored by raising the deficient nutrients. However, the effects of combined NPK fertilizer applications on soil properties and microorganisms in *Sapindus mukorossi* plantations have not been reported. The relationships between soil physicochemical properties, soil microbial attributes, and *Sapindus mukorossi* yield under different fertilization strategies are still unclear. To address these questions, we conducted a field experiment in a typical production area of *Sapindus mukorossi* in China, with the objective of exploring the following inquiries: (1) how N, P, and K fertilization would separately or combined affect the soil properties, bacterial community composition and diversity, and *Sapindus mukorossi* yield, and (2) whether fertilization could indirectly affect microbial community characteristics by altering soil properties, thereby increasing plant fruit yield.

## 2. Materials and Methods

### 2.1. Experimental Site

The research site was situated in Jianning County, Fujian Province, specifically at coordinates 116°47′20″ E and 26°40′3″ N. It experiences an average annual temperature of 17.0 °C, an average annual rainfall amounting to 1792 mm, and a high relative humidity of 84%. The soil type at the test site was sandy clay loam. The nutrient status of the soil in this experimental area was 7.75 g·kg^−1^ of soil organic carbon, 1.40 g·kg^−1^ of total nitrogen, 0.36 g·kg^−1^ of total phosphorus, 27.95 g·kg^−1^ of total potassium, 48.16 mg·kg^−1^ of fast-acting potassium, 1.32 mg·kg^−1^ of effective phosphorus, and 36.01 mg·kg^−1^ of alkaline dissolved nitrogen [8,9]. The raw material forest comprises the “Yuanhua” asexual line of Sapindales trees, featuring an average tree height of 2.38 m, an average ground diameter of 6.97 cm, and an average crown width of 2.4 m × 2.2 m (Figure 1).

### 2.2. Experimental Design

The six-year-old asexual *Sapindus mukorossi* raw material forest, known as “Yuanhua”, served as the test material, featuring a row spacing of 4 m × 4 m. Three hectares made up the entire experimental sample area. A “3414” randomized block design was adopted for the formulation of the fertilizer trial, i.e., three elements (N, P, K), four fertilization levels (0 level: 0 kg/hm^2^ of N, P, and K; 1 level: 300 kg/hm^2^ of N, 250 kg/hm^2^ of P, and 200 kg/hm^2^ of K; 2 level: 600 kg/hm^2^ of N, 500 kg/hm^2^ of P, and 400 kg/hm^2^ of K; and 3 level: 900 kg/hm^2^ of N, 750 kg/hm^2^ of P, and 600 kg/hm^2^ of K), 14 treatments, three replications, and a total of 42 treatment plots. Fertilizer application rates and levels are based on the results of our team’s previous research and the preliminary experiments we conducted, combined with the soil background data of our experimental area to derive the fertilizer application rates for this experiment. The fertilizer application rates are outlined in Table 1, with isolation rows established between the various treatment plots. Fertilizers were applied on 10 April 2021 (flowering fertilizer, 30% of the total), 20 July 2021 (fruit-strengthening fertilizer, 30% of the total), and 1 November 2021 (post-harvest fertilizer, 40% of the total) [8,9], using the furrow method of fertilizer application and mulching the soil immediately after mixing and applying the fertilizers according to the amount of fertilizer, and the remaining maintenance and management measures were identical to those of the control group. The test fertilizers were potassium sulfate (containing K_2_O 60.0%) as the sole source of K, calcium sulfate (containing P_2_O_5_ 12.0%) as the sole source of P, and urea (containing N 46.0%) as the sole source of N.

Soil sampling: In 2021, during the expansion stage of *Sapindus mukorossi*, each plot adopted the “five-point sampling method” (sampling points were evenly distributed at regular intervals (equidistant) within the area), digging 0–20 cm soil profile at each sampling point for sampling, and mixing uniformly on a plot-by-plot basis. The mixed soil samples were split into two sections. One section, after being air-dried, finely ground, and sieved to 2 mm, was transported back to the laboratory for the assessment of soil chemical properties. In order to determine the soil microorganisms, the remaining soil samples were separated from contaminants and root residues, placed in sterile bags, and then promptly transported to the laboratory under low-temperature conditions to be kept in the freezer at −80 °C.

### 2.3. Soil Physical and Chemical Properties and Yield Determination

Using a 2300 Kjeltec analyzer (FOSS, Sweden), the Kjeldahl nitrogen determination method was used to measure the total nitrogen of the soil, while the potassium dichromate oxidation-external heating method was applied to determine the organic carbon of the soil [27]. Alkaline dissolution diffusion was used to determine the amount of available nitrogen (AN); the molybdenum-antimony antimicrobial colorimetric method was used to determine the amount of total and available phosphorus in the soil; atomic absorption was used to determine the amount of potassium in the soil; flame photometry was used to determine the amount of quick potassium; and a pH meter was used to measure the pH of the soil in a mixture of soil and water (1:2.5).Yield measurement: Yield measurements for *Sapindus mukorossi* were conducted on 5 November 2021, at the point of fruit maturity during harvest. All fruits were ripe at the time of harvesting the plants. To determine the fruit yield for each plot, all ripe fruits were collected from each entire tree, weighed using an electronic scale post-harvest, and subsequently, the yields per tree were summed for each respective plot.

### 2.4. DNA Extraction, PCR Amplification and Sequencing

Total DNA was isolated from 0.5 g of soil samples using the Soil Rapid DNA SPIN Kit (Q-BIO gene, Irvine, CA, USA), and DNA quality and concentration were assessed by agarose gel electrophoresis and Invitrogen (Carlsbad, CA, USA). The bacterial 16S rRNA V3–V4 region was amplified using primers 338F (50-ACTCCTACGGGGAG GCAGCA-30) and 806 (50-GGACTACHVGGGGTWTCTAAT-30) [2,28]. PCR amplification was performed in a total reaction volume of 10 µL containing DNA template (5–50 ng), Vn F (10 µM) 0.3 µL, Vn R (10 µM) 0.3 µL, KOD FX Neo Buffer 5 µL, dNTP (2 mM) 2 µL, KOD FX-Neo 0.2 µL, ddH2O to 10 µL. PCR amplification conditions were: initial denaturation at 95 °C for 5 min, followed by denaturation at 95 °C for 30 s, annealing at 55 °C for 20 s and extension at 72 °C for 30 s for a total of 30 cycles, and finally extension at 72 °C for 10 min. PCR amplicons were purified with Agencourt AMPure XP Beads (Beckman Coulter, Indianapolis, IN, USA). Finally, these PCR products were sent to Biomark Technologies (Beijing, China) for sequencing.

### 2.5. Sequence Splicing and Annotation

The raw sequences underwent quality filtering and merging processes, facilitated by Fastp (v.0.19.6) and FLASH (v.1.2.11), respectively. Subsequently, operational taxonomic unit (OTU) clustering and chimera removal were executed using Upraise software (v.11), adhering to a 97% similarity threshold. Taxonomic assignment of 16S rRNA sequence reads was determined using the bacterial SILVA reference database (Release 138) and the Unite reference database (Release 8.0), respectively. For both databases, the RDP classifier (version 2.11) was employed to annotate the OTU representative sequences taxonomically, with a confidence threshold of 0.7 set to ensure the accuracy of the taxonomic annotation results. The sample sequence was normalized based on the minimum sequence count to generate standardized data, which were then used to calculate the Shannon index and Chao index, following the methodology outlined by Liu et al. [27], and compositions of the soil microbial community were analyzed at the phylum level.

Simpson’s diversity and Shannon’s diversity index were calculated. It is defined as: *D* and $H^{\prime }$(1)$$D=1-\sum_{i=1}^{S}P_{i}^{2}$$(2)$$H^{\prime }=-\sum_{i=1}^{S}p_{i}\ln P_{i}$$

Equations (1) and (2): $P_{i}$ denotes the relative abundance of species inside the ith soil.

### 2.6. Data Processing and Analysis

One-way analysis of variance (ANOVA) and multiple range tests for least significant difference were performed to assess the effects of different NPK fertilization on soil properties, indices of soil bacterial community diversity (OTU richness, Simpson, and Shannon indices), soil microbial community composition, and the relative abundance of genes associated with different functional classes. The differences in bacterial community distribution were analyzed by PCoA using the BMK Cloud (www.biocloud.net 25 November 2022) platform. Redundancy analysis (RDA) and mapping of bacterial community-soil property relationships were performed using Canoco 5 software. Heat maps of correlation between bacterial dominant phyla and soil environmental factors were plotted using R 4.2.0, and partial least squares path modeling (PLS-PM) was used to investigate the possible causal relationships between nitrogen, phosphorus, and potassium fertilizer application, soil properties, soil community diversity, community structure, and saprotrophic yield. We used 999 bootstraps to calculate the coefficient of determination (R^2^) and the path coefficient. The variance explained by the model is represented by the R^2^ value. The goodness of fit indicates the best overall predictive performance. Path coefficients provide information on the direction and magnitude of the relationship between variables. Indirect effects can be determined by multiplying the path coefficients between two variables [29].

## 3. Results

### 3.1. Effect of Fertilization on Soil Properties

The impact of various fertilization treatments on the soil characteristics of *Sapindus mukorossi* varied significantly, as Table 2 illustrates. The N_2_P_2_K_2_ treatments resulted in a considerable increase in soil total nitrogen, total potassium, organic carbon, available nitrogen, available phosphorus, and quick-acting potassium contents when compared to the control (Table 2). Soil pH was considerably lowered by increasing N or P treatment at different levels of nitrogen and phosphorus addition, respectively. After fertilization, the soil’s nutrient contents all showed an upward trend when compared to the control. The N_2_P_2_K_2_ and N_3_P_2_K_2_ treatments had comparatively higher total and available nitrogen contents, which were 1.4 times higher than those of the control. The N_1_P_2_K_1_ treatment had the greatest total phosphorus concentration, 2.5 times that of the control. The N_2_P_3_K_2_ and N_2_P_2_K_2_ treatments had comparatively high available phosphorus contents, which were 5.1 and 3.8 times the control, respectively. The N_2_P_2_K_3_ treatment had comparatively high total and quick potassium concentrations, which were 1.3 and 1.4 times greater than the control, respectively. The N_1_P_1_K_2_ treatment had the highest organic carbon content, 1.5 times that of the control. Each of these fertilization treatments differed from the control in a significant way (*p* < 0.05).

### 3.2. Impact of Fertilization on Soil Bacterial Community Structure and Diversity

Soil bacterial community structure produced changes under different fertilization treatments. Based on the composition of the top 10 species of bacterial community distribution in the samples analyzed at the phylum level (Figure 2a), the dominant phyla in the different treatments, except for unclassified bacteria, were Proteobacteria, Acidobacteriota, Firmicutes, Bacteroidota, Actinobacteriota, Green Benders (Chloroflexi), Verrucomicrobiota, Planctomycetota and Fusobacteriota, with a combined relative abundance of 91.89–97.09%. Among them, the relative abundance of Proteobacteria was the highest in the N_1_P_1_K_2_ treatment and significantly different from all other treatments (*p* < 0.05); Bacteroidota was relatively high in the N_2_P_3_K_2_ and N_1_P_1_K_2_ treatments and significantly different from all other treatments except N_2_P_1_K_1_ (*p* < 0.05). At different levels of nitrogen addition, the highest relative abundance of Acidobacteriota and Verrucomicrobiota microbiota was found in the N_1_P_2_K_2_ treatment, the highest relative abundance of Chloroflexi was found in the N_2_P_2_K_2_ treatment, and the lowest relative abundance of green Chloroflexi was found in the N_3_P_2_K_2_ treatment, with a significant difference between the two treatments as the nitrogen fertiliser application increased (*p* < 0.05).

Principal component analysis based on different fertilization treatments at the OTU level showed (Figure 2b) that the total variance explained by the first two axes (PC_1_ and PC_2_) of the bacterial community under different fertilization treatments was 18.72% and 12.65%, respectively, with a cumulative contribution of 31.37% (Figure 2b). In the first principal component, the bacterial community of N_1_P_1_K_2_ and N_1_P_2_K_1_ treatments differed the most, and in the second principal component, the bacterial community of N_1_P_2_K_2_ and N_3_P_2_K_2_ treatments differed the most.

Various fertilization methods exerted specific influences on bacterial α-diversity (Table 3). Bacterial community diversity demonstrated a pattern of initial increase followed by a decline in response to increasing levels of nitrogen, phosphorus, and potassium fertilization; the richness index and Shannon index of N_3_P_2_K_2_ and N_1_P_1_K_2_ were at a lower level and differed significantly from the other treatments (*p* < 0.05), and the Shannon index of N_1_P_2_K_2_, N_2_P_0_K_2_, and N_2_P_2_K_2_ treatments was notably elevated compared to all other treatments.

### 3.3. Analysis of Sapindus mukorossi Yield

Effects of various fertilization treatments with nitrogen, phosphorus, and potassium on *Sapindus mukorossi* yield and financial gains (Figure 3). The yield can be significantly increased by fertilization, which also enhances the financial gains. While the yield of nutrient deficiency treatment was significantly higher than that of total nutrient treatment and significantly lower than that of fertilization treatment, N_2_P_2_K_2_ was the highest yield compared to both fertilization and nutrient deficiency treatments. The yield of low, medium, and high nitrogen fertilizer treatments improved by 51%, 85%, and 61%, respectively, at different levels of nitrogen addition, while the economic revenue increased by 7275, 12,150, and 8700 yuan/hm^2^. At different levels of alkaline addition, compared with no phosphate fertilizer, the low, medium, and high phosphate fertilizer treatments increased the yield by 67%, 85%, and 45% and increased the economic income by 9376, 12,150, and 6362 yuan/hm^2^. At different levels of phosphate addition, the low, medium, and high phosphate fertilizer treatments raised the yield by 67%, 85%, and 45% and the economic revenue by 9376, 12,150, and 6362 yuan/hm^2^, respectively, in comparison to no phosphate fertilizer. When low, medium, and high potassium fertilizers were applied at different levels of alkaline addition, the yield and economic income increased by 45%, 85%, and 20%, respectively, and by 7275, 12,150, and 3263 yuan/hm^2^, in comparison to no potassium fertilizer. Balanced application of nitrogen, phosphorus, and potassium fertilizer significantly impacts yield formation.

### 3.4. Relationships Between Bacterial Communities, Sapindus mukorossi Yield and Soil Properties

There were correlations between bacterial community α-diversity (species richness, Simpson and Shannon indices), yield, and soil environmental factors (Table 4). Soil pH and total phosphorus content were significant factors (*p* < 0.05) affecting bacterial community α-diversity, where soil pH was positively correlated with species richness, Simpson, and Shannon indices, while total phosphorus was negatively correlated with all three. Except for total potassium and quick-acting potassium content, all other soil property indicators were significantly correlated (*p* < 0.05) with fruit yield of *Sapindus mukorossi*, where pH was negatively correlated with yield and the rest of the nutrient contents were positively correlated with yield.

The relationship between bacterial community and soil physicochemical properties at the portal level was studied and analyzed by RDA, and the first axis explained 36.07% of the bacterial community, the second axis explained 7.54% of the bacterial community, and the two axes together explained 43.61% of the bacterial community (Figure 4a). On the first axis, the factors with significant correlation with bacterial community structure were pH, total potassium, organic carbon, total phosphorus and available phosphorus. The correlation between each soil factor and bacterial community structure on the second axis was not significant. The relative abundance of *Acidobacteriota*, *Verrucomicrobiota*, *Actinobacteriota*, *Planctomycetota* and *Chloroflexi* in the bacterial community were all positively correlated with soil pH. The relative abundance of *Firmicutes*, *Fusobacteriota* and *Bacteroidota* decreased with increasing pH, but exhibited a positive correlation with other soil environmental factors. As shown in Figure 4b, the soil environmental factors of pH, total potassium, and available phosphorus content all had a significant effect on the dominant phylum of soil microbial bacteria (*p* < 0.05).

By correlating the relative abundance of the dominant phyla with soil environmental factors (Figure 5), it could be learned that *Fusobacteriota* and *Firmicutes* were all significantly negatively correlated with soil pH (*p* < 0.05), whereas there was a significant positive correlation (*p* < 0.05) with the contents of total phosphorus and available phosphorus (*p* < 0.05). Among them, *Firmicutes* and *Bacteroidota* were also significantly (*p* < 0.05) positively correlated with organic carbon content. Both *Acidobacteriota* and *Verrucomicrobiota* were significantly (*p* < 0.05) negatively correlated with total phosphorus, total potassium, available phosphorus, and organic carbon content. In addition, both *Acidobacteriota* and *Chloroflexi* showed a statistically significant positive correlation with soil pH (*p* < 0.05).

### 3.5. Partial Least Squares Path Model (PLS-PM) Analysis

The relationship between NPK fertilization on *Sapindus mukorossi* yield, soil properties, soil community diversity, and community structure was analyzed based on PLS-PM (Figure 6a). In this model predicting *Sapindus mukorossi* yield, the variance attributed to soil properties was 0.67, soil bacterial diversity was 0.21, bacterial community structure was 0.57, and the goodness-of-fit index was 0.54 (Figure 6a). Fertilization had a direct positive effect on soil properties (0.82), as well as on soil bacterial community diversity (0.30) and community structure (0.002). Meanwhile, soil properties had a direct significant effect (0.93) on yield, while community diversity (0.14) and community structure (−0.05) contributed less to the variation in *Sapindus mukorossi* yield. In addition, the indirect effect of nitrogen, phosphorus, and potassium fertilization on *Sapindus mukorossi* yield was 0.71, while the direct effect by soil properties was 0.93 (Figure 6b). Overall, soil nutrient content was the direct driver of *Sapindus mukorossi* yield, while bacterial community diversity and structure did not significantly affect yield.

## 4. Discussion

### 4.1. Effect of Fertilization on Soil Properties

The study revealed that augmenting nitrogen application on the basis of P_2_K_2_ and phosphorus application on the basis of N_2_K_2_ notably decreased soil pH, likely due to the urea treatment exacerbating soil acidification [30]. During the oxidation process of NH^4+^, H^+^ is released back into the soil, exchanging cations that are absorbed by plants, thereby accelerating soil acidification [30,31,32]. N_2_P_2_K_2_ fertilization treatment is more efficient in increasing soil nutrient content, especially the content of soil available nutrients, which is at a lower level when the nitrogen application is too low or too high. A certain degree of N addition has a positive effect on soil fertilization and can accelerate the decomposition of soil organic matter [33,34,35]. Some studies have shown that the application of N, P, and K in pairs or individually can promote plant root growth and increase the input of inter-root organic matter, which in turn increases the organic carbon content [36,37]. However, excessive fertilizer application resulted in a decrease in the overall amount of organic carbon, which could be because excessive fertilizer application can speed up the biodegradation of organic carbon sources by altering the quantity and activity of soil microorganisms [35]. In this study, it was found that the soil total and available nitrogen contents showed an increase and then a decrease trend with the increase in nitrogen application, which was inconsistent with the results of Qiao et al. [36] and could be explained by the theory of “stoichiometric decomposition” proposed by Hessen et al. [38]. The stoichiometric imbalance after adding nitrogen has been proven to accelerate the decomposition of soil organic carbon to maintain soil carbon/nitrogen ratios, with decomposition rates and microbial activity being highest when carbon and nitrogen inputs corresponded to microbial stoichiometric carbon to nitrogen ratios [39,40]. In a previous study conducted in energy forests, N additions accompanied by meeting the needs of microbial growth and stimulating the mineralization of native soil organic matter as well as the release of reactive N during nitrification, denitrification, and leaching [31]. This highlights the importance of rational application of nitrogen fertilizers.

### 4.2. Effect of Fertilizer Application on Soil Bacterial Communities

Soil microorganisms play an important role in both energy flow and nutrient cycling processes in the soil, and the richer the soil microbial species, the greater the plant resistance to diseases [41,42]. Meanwhile, soil microorganisms can reflect soil fertility and its quality status, and changes in their diversity and activity can also indicate soil health [43]. However, in this study, it was found that the application of nitrogen fertilizer at low versus moderate levels was beneficial in increasing the bacterial community diversity, while excessive nitrogen additions instead decreased the bacterial community diversity. Consistent with the results of most previous studies, N addition could significantly affect bacterial community diversity, but excessive nitrogen fertilization reduced the number of species in the soil microbial community, which may be due to the fact that the high nitrogen environment is more favorable for some nitrogen-loving microorganisms, but their microbial metabolisms still showed a general decreasing trend [44,45]. The reduction in species diversity resulting from nitrogen enrichment poses a risk to ecosystem stability and can disrupt interactions between above- and below-ground ecosystems. Furthermore, phosphorus addition not only boosts microbial biomass but also modifies the makeup of soil microbial communities [46]. We further found that nitrogen addition and phosphorus addition led to significant changes in the diversity of bacterial communities in the soil, while there was no significant difference in the effect of potassium fertilization on bacterial community diversity (Table 3). Our results were consistent with those of previous authors [14,21], with nitrogen deficiency having the greatest effect on bacterial community structure and composition, phosphorus deficiency having the greatest effect on network construction and bacterial activity, and potassium deficiency having the least effect. In our experiment, bacterial community diversity showed an increasing and then decreasing trend with increasing N, P, and K fertilization (Table 3), suggesting that lower nutrient inputs may lead to higher plant abundance, which may result in a homogenizing effect on the soil bacterial community, and vice versa. Some studies have found that the application of N, P, and K fertilizers had no or a positive effect on bacterial communities [45,47,48]. The inconsistent results of these studies may be due to different durations and doses of applied N [42,49].

In addition, the reduction in bacterial diversity may also be related to host plant characteristics, as the organs of *Sapindus mukorossi* have a certain saponin content [50,51]. which has strong bacteriostatic activity, and it has been noted that it significantly inhibits the growth of *Staphylococcus aureus* [51], *Bacillus subtilis* [50], *Bacillus cereus* [52], *Escherichia coli* [50,53], and other microbial taxa. Our previous study found that fertilization significantly increased the saponin content of *Sapindus mukorossi*, further confirming that saponin may be a potential factor affecting bacterial activity [53]. In addition, in the process of *Sapindus mukorossi* cultivation, weeds and pests are easy to grow, and inevitably a certain amount of pesticides needs to be sprayed, which may indirectly affect the root secretion or directly inhibit the reproduction of certain microorganisms in the soil [54,55].

Fertilization, herbicides, and external environmental changes are the main factors that hinder natural selection in cash crops [54]. Fertilization significantly affected soil microbial community composition [56]. In this study, the first dominant phylum was *Ascomycetes*, which is mainly involved in soil nutrient regulation and has a major impact on soil nutrient cycling, although *Ascomycetes* varied with different levels of fertilization and crop selection [54]. Our finding aligns with previous studies that fertilization increases the relative abundance of *Ascomycetes* [44,45]. The relative abundance of *Acidobacteriota,* a typical nutrient-poor taxon, was highest in the N_1_P_2_K_2_ treatment [57]. The organic carbon pool in the soil can be increased by fertilizer application to give the soil a higher nutrient affinity, and high soil fertility can drive the shift from nutrient-poor to eutrophic taxa in the bacterial community, resulting in superior soil conditions, which consequently increased community diversity and intensified competition [58]. From the perspective of fertilization, the increase in nitrogen application can change the soil nitrogen content and thereafter affect the distribution of the soil bacterial community or indirectly affect the structure of the soil bacterial community by changing the pH value of the soil, the content of organic carbon, and the soil carbon and nitrogen ratio, etc. [49,58,59].

### 4.3. Soil Bacterial Communities in Relation to Environmental Factors

To date, there is sufficient evidence that soil physicochemical factors such as SOC, TP, TN, AP, AN, and AK are enhanced by bacterial diversity, while other fertilizers can ameliorate soil acidification to some extent [60]. According to the results of RDA analysis, soil environmental factors play a major role in microbial community structure, and this result has been confirmed by a large number of studies [18,61]. We found that soil pH, available phosphorus, and total potassium were the main environmental factors affecting the distribution of bacterial microbial communities, accounting for 80% of the variation in the composition of community structure (Figure 4b). It is generally accepted that soil pH is often considered the main factor regulating soil bacterial communities, and bacteria are very sensitive to soil pH, with lower pH soils being accompanied by aluminum toxicity and nutrient leaching, which negatively affects bacterial growth [62]. Microbial communities typically thrive more vigorously in neutral or slightly alkaline environments [62], whereas they tend to grow poorly in conditions of low pH [63]. This may be due to the fact that a strongly acidic environment inhibits enzyme activity and whole cell metabolism [62,64]. This is in agreement with our results that the number of *Acidobacteriota* and *Chloroflexi* were significantly and positively correlated with pH. In addition, changes in soil pH were also produced during fertilization, suggesting that different fertilization treatments can alter soil bacterial habitat conditions, thus causing changes in soil microbial community structure. For example, fertilizer application caused a decrease in alkaline cations and an increase in acidic cations. This is consistent with the observations of Chen et al. [65].

Available phosphorus in soil has a significant effect on bacterial communities. Phosphorus is essential for the synthesis of RNA, DNA, and ATP, and the source of nutrients plays a crucial role in bacterial growth [19]. Soil microorganisms are key regulators of soil phosphorus mobilization and transformation, and they enhance soil phosphorus effectiveness through inorganic phosphorus solubilization and organic phosphorus mineralization through the regulation of phosphorus cycling genes and phosphatase expression [18]. Fertilizer management can provide a suitable environment for soil microbial growth by stimulating microbial activity [66,67]. We found that the numbers of *Fusobacteriota*, *Firmicutes,* and *Bacteroidota* were significantly and positively correlated with total and available phosphorus after fertilization. This indicates that different fertilization treatments cause changes in microbial nutrient sources, which in turn affects the distribution of bacterial communities. This is further evidenced by the fact that balanced application of NPK improves soil fertility while promoting soil ecosystem stability and health. Therefore, fertilizer application and soil microbial activity are frequently studied in these studies, but there is still a need to assess the optimal application rate of NPK fertilizers.

### 4.4. Fertilizer Application Affects the Sapindus mukorossi Yield by Influencing Soil Properties Rather Bacteria

Soil microbial diversity is positively correlated with biotic and abiotic stress resistance and nutrient cycling rates, as well as increasing plant productivity through selective effects or complementarity, but soil properties and plant production do not always increase with increased microbial community diversity and structure [68]. After fertilization, fertilizers may be absorbed and utilized by plants or remain in the soil, leading to changes in protists [69], bacteria [70], and fungal communities [59]. Therefore, the increase in plant yield after fertilization may be related to the physical, chemical, and biological properties of the soil. According to PLS-PM analysis in this study, NPK fertilization had a direct effect on *Sapindus mukorossi* yield with a direct path coefficient of −0.07, which could be attributed to the fact that fertilization provided essential nutrients for *Sapindus mukorossi* growth. However, the path coefficient of the indirect effect of fertilizer application on *Sapindus mukorossi* yield (0.74) was significantly higher than the direct effect (Figure 6b), mainly due to nutrient variation in soil properties (0.93) (Figure 6a), suggesting a preliminary role of soil nutrient content in driving the yield. More importantly, nitrogen, phosphorus, and pH in the soil were the most important key factors affecting the yield (Table 4). This is similar to the results of Lv et al. [16], where phosphorus level in the soil was the limiting factor for crop yield. Soil nutrient levels, especially total nutrient levels, are the main cause of variation in crop yield, while bacteria indirectly affect yield through enzyme activities and nutrient levels [71]. In this study, high nitrogen, phosphorus, and potassium treatments resulted in lower pH and caused soil acidification, hindering bacterial diversity. This is due to the fact that fertilizer application provides the N, P, and K sources needed for soil microbial growth; therefore, the change in soil pH caused by nitrogen, phosphorus, and potassium application is the main reason for the diversity of microbial communities. Furthermore, excessive fertilizer application can cause significant soil erosion, disrupting the balance of soil nutrient structure. This imbalance, in turn, hinders the growth and development of plant roots, diminishes their capacity to absorb nutrients and water, and ultimately results in decreased crop yields [21]. Fertilization also had a direct effect on soil bacterial diversity with a path coefficient of 0.30, and soil properties had a significant direct effect on soil microbial diversity with a path coefficient of −0.66, suggesting that a shift in the diversity of microbial communities was driven by the soil environmental factors under the different fertilization treatments.

Interestingly, according to the PLS-PM results, soil microbial diversity and community structure had little direct effect on *Sapindus mukorossi* yield, with direct path coefficients of 0.15 and −0.05, respectively (Figure 6a), which is in agreement with the results of Ma et al. [39]. This further confirmed the fact that crop growth was mainly influenced by soil fertility. Therefore, soil nutrients become a key environmental factor influencing microbial community succession and yield enhancement [59,68,72], and a balanced fertilization pattern provides a relatively balanced nutrient environment. Our results suggest that the N_2_P_2_K_2_ treatment maximizes *Sapindus mukorossi* yield and maintains soil health and bacterial microbial diversity.

## 5. Conclusions

In this study, we investigated the response of soil nutrients and bacterial communities to balanced fertilization with nitrogen, phosphorus, and potassium and their correlations with the yield of *Sapindus mukorossi* in raw material forests of *Sapindus mukorossi*. Different fertilization treatments had significant effects on soil acidity (pH) and nutrient content. The addition of nitrogen or phosphorus led to an increase in the degree of soil acidification, which would lead to the accelerated decomposition of organic carbon in the soil. Fertilization with combinations of nitrogen (N), phosphorus (P), and potassium (K) can significantly change soil bacterial community structure and improve microbial diversity in *Sapindus mukorossi* forests by changing soil properties, but the changes in bacterial community structure and diversity did not significantly affect *Sapindus mukorossi* yield. The primary determinants influencing *Sapindus mukorossi* output under fertilization treatments were variations in soil nutrients, particularly N and P concentrations. However, we believe that more research is necessary to examine how the fertilization treatments affect the functions of the microbial community and how this can affect future *Sapindus mukorossi* yield. In order to increase soil productivity and lower environmental pollution, scientific fertilizer application strategies should be established, given that soil parameters have a substantial impact on the structure of bacterial communities. Meanwhile the research holds broader implications for sustainable agriculture and soil health management.

## Figures and Tables

**Figure 1 microorganisms-13-00173-f001:**
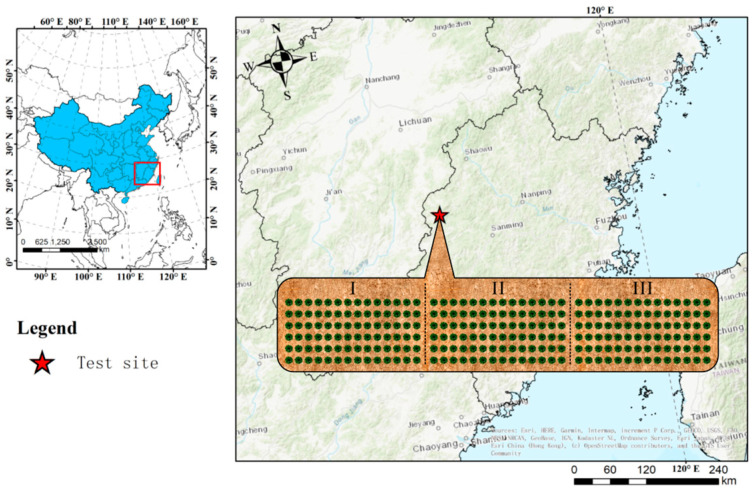
Location of the experimental site and experimental design of different NPK treatments.

**Figure 2 microorganisms-13-00173-f002:**
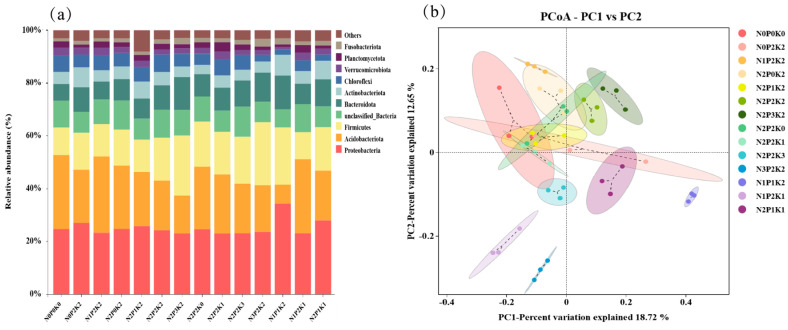
Relative abundance of horizontal species of soil bacteria under different fertilization treatments. (**a**) Relative abundance of portal bacterial level species. (**b**) Bacterial OTU horizontal principal component analysis.

**Figure 3 microorganisms-13-00173-f003:**
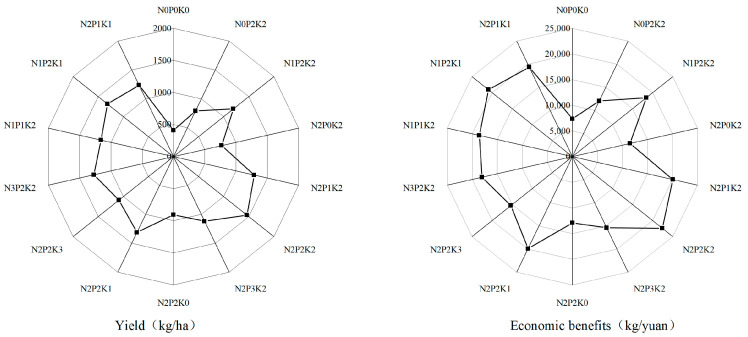
Yield and economic benefit of *Sapindus mukorossi* under different fertilization treatments.

**Figure 4 microorganisms-13-00173-f004:**
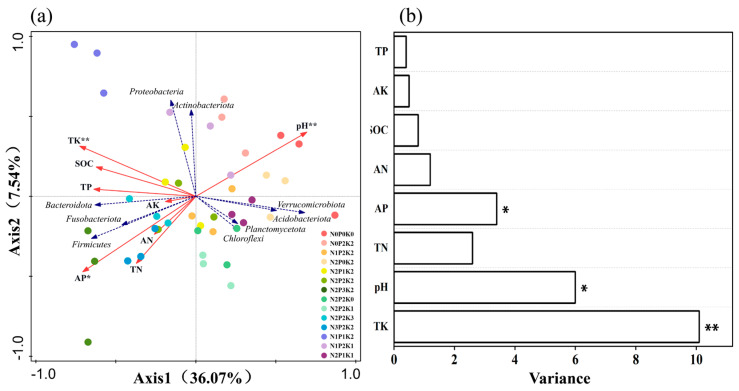
Biplot of the RDA analysis of soil bacterial communities under different fertilization treatments. (**a**) RDA analysis of bacterial communities. (**b**) Soil environmental factor parameter variables. Note: * is significant at the 0.05 level and ** is significant at the 0.01 level.

**Figure 5 microorganisms-13-00173-f005:**
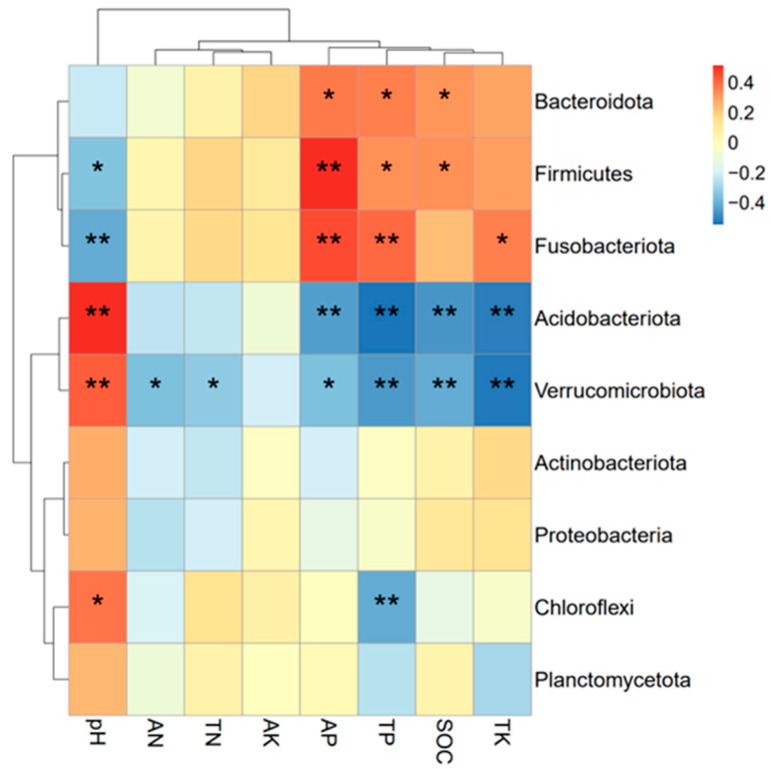
Correlation analysis between dominant bacteria and soil environmental factors. Note: * is significant at the 0.05 level and ** is significant at the 0.01 level.

**Figure 6 microorganisms-13-00173-f006:**
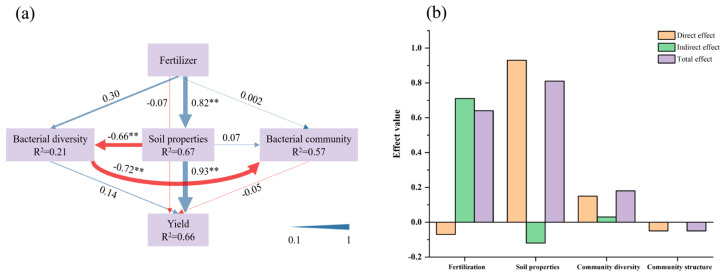
(**a**) Direct and indirect effects of fertilizer, soil properties, and bacterial attributes on the *Sapindus mukorossi* yield. (**b**) Overall effects of fertilization, *Sapindus mukorossi* soil properties, community diversity, and structure on yield. Note: Soil properties refer to total nitrogen, available nitrogen, total phosphorus, available phosphorus, total potassium, quick-acting potassium, organic carbon content, and pH; bacterial diversity is OTU richness, Shannon index, and Simpson index; and community structure is the first two axes of principal components (PC_1_ and PC_2_). The numbers on the arrows are standardized path coefficients, with blue arrows indicating positive effects and red arrows indicating negative effects. Path coefficients and coefficients of determination (R^2^) were derived after 999 bootstrapping attempts. R^2^ values indicate the variance explained by the model. The model was evaluated using the goodness-of-fit statistic, which is a measure of overall predictive performance. Significance is indicated by different path coefficients, ** *p* < 0.01.

**Table 1 microorganisms-13-00173-t001:** Factor-level combinations in each treatment.

Processing Number	Fertilization Level (kg·hm^−2^)		
	N	P	K
N_0_P_0_K_0_	0	0	0
N_0_P_2_K_2_	0	500	400
N_1_P_2_K_2_	300	500	400
N_2_P_0_K_2_	600	0	400
N_2_P_1_K_2_	600	250	400
N_2_P_2_K_2_	600	500	400
N_2_P_3_K_2_	600	750	400
N_2_P_2_K_0_	600	500	0
N_2_P_2_K_1_	600	500	200
N_2_P_2_K_3_	600	500	600
N_3_P_2_K_2_	900	500	400
N_1_P_1_K_2_	300	250	400
N_1_P_2_K_1_	300	500	200
N_2_P_1_K_1_	600	250	200

**Table 2 microorganisms-13-00173-t002:** Changes in soil properties under different fertilization treatments.

	pH	TN (g·kg^−1^)	TP (g·kg^−1^)	TK (g·kg^−1^)	SOC (g·kg^−1^)	AN (mg·kg^−1^)	AP (mg·kg^−1^)	AK (mg·kg^−1^)
N_0_P_0_K_0_	5.02 ± 0.04 a	1.28 ± 0.08 i	0.42 ± 0.01 i	22.95 ± 0.69 h	11.08 ± 0.75 h	90.21 ± 6.51 f	1.13 ± 0.07 h	81.79 ± 2.90 g
N_0_P_2_K_2_	5.09 ± 0.02 a	1.33 ± 0.06 hi	0.46 ± 0.03 i	26.58 ± 0.57 de	12.76 ± 0.53 fg	93.49 ± 0.99 f	2.47 ± 0.14 g	93.57 ± 1.25 cdef
N_1_P_2_K_2_	4.82 ± 0.07 c	1.51 ± 0.05 def	0.52 ± 0.05 hi	26.27 ± 0.32 e	13.74 ± 0.81 def	105.59 ± 3.82 cd	2.97 ± 0.08 g	104.36 ± 1.54 abc
N_2_P_0_K_2_	5.04 ± 0.02 a	1.53 ± 0.02 de	0.49 ± 0.06 hi	24.92 ± 0.45 fg	13.52 ± 0.68 defg	97.36 ± 4.17 ef	2.80 ± 0.45 g	92.68 ± 4.55 defg
N_2_P_1_K_2_	4.84 ± 0.07 c	1.64 ± 0.03 bc	0.61 ± 0.05 g	27.72 ± 0.66 cd	14.21 ± 0.67 d	113.92 ± 5.86 bc	4.32 ± 0.58 cde	86.17 ± 9.31 fg
N_2_P_2_K_2_	4.82 ± 0.03 cd	1.79 ± 0.04 a	0.87 ± 0.04 bcd	28.26 ± 0.93 bc	15.41 ± 0.68 bc	126.66 ± 5.42 a	5.29 ± 0.22 ab	107.46 ± 6.89 ab
N_2_P_3_K_2_	4.67 ± 0.05 f	1.49 ± 0.06 def	0.96 ± 0.02 b	25.79 ± 0.43 ef	14.05 ± 0.78 de	95.77 ± 5.23 f	5.73 ± 0.26 a	88.81 ± 3.15 efg
N_2_P_2_K_0_	4.91 ± 0.03 b	1.44 ± 0.06 efg	0.63 ± 0.03 fg	24.76 ± 0.81 fg	12.99 ± 0.37 efg	109.63 ± 6.12 cd	4.46 ± 0.29 cd	96.26 ± 4.52 cdef
N_2_P_2_K_1_	4.81 ± 0.04 cd	1.60 ± 0.01 cd	0.56 ± 0.06 gh	24.12 ± 0.91 gh	15.68 ± 0.47 b	106.3 ± 1.39 cd	4.78 ± 0.15 bc	90.53 ± 7.27 efg
N_2_P_2_K_3_	4.71 ± 0.00 ef	1.67 ± 0.09 bc	0.72 ± 0.05 ef	28.97 ± 0.76 ab	14.38 ± 0.34 cd	115.11 ± 3.88 b	3.84 ± 0.39 ef	114.26 ± 10.78 a
N_3_P_2_K_2_	4.56 ± 0.04 h	1.74 ± 0.03 ab	0.80 ± 0.02 de	27.65 ± 0.77 cd	13.85 ± 0.67 de	123.92 ± 2.97 a	4.85 ± 0.29 bc	87.99 ± 4.36 efg
N_1_P_1_K_2_	4.78 ± 0.04 cde	1.47 ± 0.06 ef	0.92 ± 0.06 bc	29.78 ± 0.38 a	16.90 ± 0.29 a	103.93 ± 2.64 de	3.96 ± 0.29 def	98.40 ± 6.33 bcde
N_1_P_2_K_1_	4.80 ± 0.03 cd	1.42 ± 0.09 fgh	1.05 ± 0.09 a	24.86 ± 0.67 fg	14.46 ± 0.28 cd	111.74 ± 6.49 bcd	3.57 ± 0.16 f	94.70 ± 6.86 cdef
N_2_P_1_K_1_	4.74 ± 0.02 def	1.36 ± 0.03 ghi	0.84 ± 0.02 cd	23.27 ± 0.49 h	12.65 ± 0.41 g	112.85 ± 4.27 bc	2.77 ± 0.26 g	101.91 ± 3.44 bcd

Note: SOC: organic carbon; TN: total nitrogen; TP: total phosphorus; TK: total potassium; AN: alkaline nitrogen; AP: active phosphorus; AK: acute potassium; pH: soil acidity and alkalinity. Different letters in the same indicator indicate significant differences between treatments (*p* < 0.05).

**Table 3 microorganisms-13-00173-t003:** Soil bacterial community diversity under different fertilization treatments.

Fertilizer Treatment	OTU Richness	Simpson	Shannon
N_0_P_0_K_0_	617.38 ± 18.54 abc	0.9954 ± 0.0004 a	8.5353 ± 0.0666 ab
N_0_P_2_K_2_	591.33 ± 47.04 bcd	0.995 ± 0.0012 a	8.4455 ± 0.2553 ab
N_1_P_2_K_2_	674.67 ± 25.66 a	0.9955 ± 0.0004 a	8.6310 ± 0.0846 a
N_2_P_0_K_2_	664.38 ± 6.58 ab	0.9954 ± 0.0003 a	8.6146 ± 0.0321 a
N_2_P_1_K_2_	610.33 ± 14.64 abc	0.9949 ± 0.0007 ab	8.4685 ± 0.0948 ab
N_2_P_2_K_2_	682.00 ± 21.79 a	0.9951 ± 0.0005 a	8.6451 ± 0.0716 a
N_2_P_3_K_2_	615.67 ± 22.19 abc	0.9932 ± 0.0011 b	8.3062 ± 0.1299 c
N_2_P_2_K_0_	630.67 ± 21.83 abc	0.9945 ± 0.0011 ab	8.4917 ± 0.0870 ab
N_2_P_2_K_1_	571.45 ± 37.08 cd	0.9948 ± 0.0010 ab	8.4505 ± 0.1507 ab
N_2_P_2_K_3_	619.04 ± 45.10 abc	0.9946 ± 0.0006 ab	8.4946 ± 0.1153 ab
N_3_P_2_K_2_	456.33 ± 110.35 e	0.9910 ± 0.0010 c	7.9449 ± 0.2511 d
N_1_P_1_K_2_	463.33 ± 12.70 e	0.9889 ± 0.0017 d	7.6773 ± 0.0709 e
N_1_P_2_K_1_	536.00 ± 26.15 d	0.9948 ± 0.0004 ab	8.4118 ± 0.0359 ab
N_2_P_1_K_1_	610.33 ± 14.64 abc	0.9942 ± 0.0006 ab	8.4310 ± 0.0190 ab

Note: Different letters in the same index signify statistically significant differences at *p* < 0.05.

**Table 4 microorganisms-13-00173-t004:** Relationship between bacterial community diversity, fruit yield, and soil environmental factors.

	pH	TN	TP	TK	SOC	AN	AP	AK
OTU richness	0.336 *	0.110	−0.345 *	−0.097	−0.211	−0.047	−0.109	0.288
Simpson	0.542 **	−0.062	−0.532 **	−0.248	−0.200	−0.172	−0.349 *	0.126
Shannon	0.484 **	0.044	−0.445 **	−0.149	−0.185	−0.036	−0.275	0.331 *
Yield	−0.699 **	0.645 **	0.634 **	0.405	0.699 **	0.773 **	0.707 **	0.378

Note: * indicates significant correlation at the 0.05 level; ** indicates significant correlation at the 0.01 level.

## Data Availability

The original contributions presented in the study are included in the article material, further inquiries can be directed to the corresponding authors.

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
