# Peer review of "Fertilization Induced Soil Microbial Shifts Show Minor Effects on Sapindus mukorossi Yield"

_microorganisms, 2025, doi:10.3390/microorganisms13010173_

Round 1
Reviewer 1 Report
Comments and Suggestions for Authors
Microorganisms
Manuscript Draft
Manuscript Number: 3400857
Title: Fertilization Induced Soil Microbial Shifts Show Minor Effects on Sapindus mukorossi Yield
Article Type: Research article
General Comments on MDPI Questions that Reviewers must answer:
- Is the manuscript clear, relevant for the field and presented in a well-structured manner?
This manuscript is written clearly, is well-structured, and is potentially relevant to the field since it focuses on the yield impacts of chemical fertilizers on crop/soil microbes. However, the manuscript needs to make the following FIVE general clarifications:
1) Please add a paragraph to the start of the Introduction section (citing literature where needed) stating the common name of Sapindus mukorossi and providing some background information to the reader on why this tree crop is economically important. Include the first two sentences on L56-59 in this new paragraph. There are other uses of this tree such as the fruit aside from bio-energy from the woody biomass.
2) In the Methods section, please add equations and writing describing the calculations of a) OTU richness, b) Simpson diversity index, and c) Shannon diversity index.
3) In the Methods section on L164-166, add more sentences to the paragraph describing the type of yield. Biomass? Fruit? What exactly was measured? What was the date(s) of harvest? Yield as is or dry matter? This is just 1.5 sentences and needs to be a solid paragraph that is a lot longer and more in-depth since the reader would not be able to replicate the experiment based on this description.
4) For the data presented in tables and figures, please do NOT use abbreviations. Understanding of tables and figures are “stand alone” and are independent of the writing. Please write out what the abbreviations mean. If you have to use another row in the header row of tables that is OK.
5) There are A LOT of formatting errors and some English errors (see below). For example, one type of error that is common is the need for a space to be added.
Given the potential contribution of this research to the improving our understanding of the impact of chemical fertilizers on agro-forestry yield and soil microbial diversity, this research warrants publication. Also, please make the following minor edits and clarifications:
1) MDPI Abstracts need to be no more than 200 words.
2) On L42, change to “…was obtained. The main factors”
3) Change i.e., to e.g., since these are examples of and not another way of saying “bacterial community attributes”
4) On L52-53, the keywords need to be in alphabetical order. Please make sure the first word of the first keyword is capitalized. Please change to: Keywords: Bacterial community; diversity; fertilization; Sapindus mukorossi; soil fertility; structural distribution
5) Throughout the manuscript, please make sure there is a space before the [#]. For example on L57, change to “…areas [1-3].
6) Also, there does no space between number for [#,#]. For example on L59, change to “…energy source [4,5].” Please make similar edits throughout the manuscript.
7) In each major section of the manuscript, define abbreviations. For example on L60, change to “…bioenergy species. Nitrogen (N), phosphorus (P), and potassium (K) are massive…”
8) Unless this is at the end of a sentence with a period, there should be a blank space on both sides of the [#]. For example on L88, change to “Ma et al. [23] showed…”
9) On L118, change to “…yield and (2) whether…”
10) In Figure 1, please increase the size of both maps so the writing can be read.
11) On L147, change to “of 2 mm to be…” where you add a space.
12) In Table 1, there does not need to be so much space between rows. Delete blank rows between rows with data.
13) Write L164 as a complete sentence: “Yield measurements of Sapindus mukorossi were taken at maturity.”
14) Add L164-166 to the end of the paragraph on L163.
15) All weblinks on L183-191 need to be put in [#] format and cited as references. See MDPI Microorganisms Word template for authors for citation formatting for this (accessed 30 December 2024).
16) On L211, change to 3. Results
17) Everywhere in the manuscript, use the sub-script formatting. There will be a lot of edits to make. For example: N2P2K2
18) For Table 2, please widen columns and increase size of font for whole table.
19) For both Figure 2 and Figure 3, put one graph on top of the other and increase size of both graphs so the writing can be read.
20) Change to Figure 3. On L289 and do not abbreviate.
21) For Table 4, widen the first column so all words fit on one row.
22) Here and everywhere else, write out Fig. as Figure such as on L325 where this should be (Figure 5)
23) The Note on L353-361 should be moved to the writing.
24) In each major section of the manuscript, define abbreviations. For example on L530, change to “…combinations of nitrogen (N), phosphorus (P), and potassium (K) can significantly change…”
25) In the References, please put journal names in italics so for example Forest Ecol. Manag. on L560. Please also add DOI links at the end of each reference.
· Are the cited references mostly recent publications (within the last 5 years) and relevant? Does it include an excessive number of self-citations?
About half of the cited references have been published within the last 5 years and appear relevant to the research topic. There are no excessive self-citations.
· Is the manuscript scientifically sound and is the experimental design appropriate to test the hypothesis?
The manuscript is scientifically sound and the experimental analyses are appropriate.
· Are the manuscript’s results reproducible based on the details given in the methods section?
The manuscript’s experimental results are NOT reproducible based on what is described in 2. Materials and Methods.
· Are the figures/tables/images/schemes appropriate? Do they properly show the data? Are they easy to interpret and understand? Is the data interpreted appropriately and consistently throughout the manuscript? Please include details regarding the statistical analysis or data acquired from specific databases.
The quality of the tables and figures need to be improved (see prior edits).
· Are the conclusions consistent with the evidence and arguments presented?
The Conclusions are consistent with the evidence and arguments presented.
· Please evaluate the data availability statements to ensure it is adequate.
All Back Matter sections are fine.
Author Response
Dear Editor-in-Chief:
We have revised our previously-submitted manuscript entitled “Fertilization Induced Soil Microbial Shifts Show Minor Effects on Sapindus mukorossi Yield” (Manuscript ID: 3400875) by carefully following the comments by the three reviewers and the editor. Now, we resubmit the new manuscript based on the suggestions.
In this revised version we have responded to each of the issues raised by the reviewers to improve both the scope and expression of the work. The major changes in the current revision include:
- We have revised the introductory section as suggested by reviewer #1 to introduce the importance of background information and economic value of Sapindus spp. We have also revised the Materials and Methods section as suggested by the reviewer by revising the whole text such as blank lines and fonts, deleting the redundant text, and refining the content to make it clearer.
- We have explained the problems as suggested by reviewer #2 “why the sampling depth was 0-20 cm” and “no mention of whether the experiment was repeated over multiple years or growing seasons”.
- We have followed the recommendations of reviewer 3 on “Rationale for the choice of fertilizer trial and fertilizer design” and “Frequency of yield measurements and whether inter-annual variability was taken into account” and “Reasons for optimization of yield by N2P2K2 treatment”. and “Reasons for optimizing yield with N2P2K2 treatment” were explained.
The detailed responses to specific comments are listed below.
Yours Sincerely,
Juntao Liu
Ph.D.
Reviewer #1:
This manuscript is written clearly, is well-structured, and is potentially relevant to the field since it focuses on the yield impacts of chemical fertilizers on crop/soil microbes. However, the manuscript needs to make the following FIVE general clarifications:
Response: Thank you for your recognition to our work. We have responded to your comments very carefully. The detailed responses to specific comments are listed below.
- Please add a paragraph to the start of the Introduction section (citing literature where needed) stating the common name of Sapindus mukorossiand providing some background information to the reader on why this tree crop is economically important. Include the first two sentences on L56-59 in this new paragraph. There are other uses of this tree such as the fruit aside from bio-energy from the woody biomass.
Response: We agree with your insightful points, We have added more background information about Sapindus mukorossi in the Introduction section will help the reader to better understand the economic importance of this study. We will add relevant content at the beginning of the introduction section detailing the common name of Sapindus mukorossi and its economic importance. The specific modifications are as follows.
Currently, Sapindus mukorossi is widely used in biomedicine, bioenergy, and cosmetics (Liu et al. 2022; Wang et al. 2021; Xu et al. 2021; Zhao et al. 2019). The whole plant of Sapindus indica, including its fruits, roots, bark and leaves, has been used in traditional medicine in China (Xu et al. 2023). Its soap pods, which have a crude protein content of about 22%, and the kernels of the soap berries, which contain about 40% fatty acids, are also woody oilseed species that have been promoted for use in recent years (Xu et al. 2022).
- In the Methods section, please add equations and writing describing the calculations of a) OTU richness, b) Simpson diversity index, and c) Shannon diversity index.
Response: Thanks for your suggestion. We recognize that adding specific detailed descriptions of OTU richness, Simpson's Diversity Index, and Shannon's Diversity Index to the Methods section would improve the clarity and reproducibility of the paper. The specific modifications are as follows.
The sample sequence was flattened according to the minimum number to obtain standardized data for calculation of the Shannon index and Chao index according to Liu et al. [27], and compositions of the soil microbial community were analyzed at the phylum level.
- In the Methods section on L164-166, add more sentences to the paragraph describing the type of yield. Biomass? Fruit? What exactly was measured? What was the date(s) of harvest? Yield as is or dry matter? This is just 1.5 sentences and needs to be a solid paragraph that is a lot longer and more in-depth since the reader would not be able to replicate the experiment based on this description.
Response: Thank you for your insightful comments. We fully agree that providing a more detailed description of the method is essential for the reproducibility of the experiment. We will expand on this paragraph in the Methods section.
Yield measurement: Sapindus niloticus was harvested at fruit maturity (November 5, 2021), and waited until all the fruits were ripe for picking the fruits of the whole plant, while Sapindus niloticus single-plant yield refers to the fruit yield of each plot by picking all the fruits that have matured from the whole tree, and then weighed the fruits using an electronic scale after picking, and then counted.
- For the data presented in tables and figures, please do NOT use abbreviations. Understanding of tables and figures are “stand alone” and are independent of the writing. Please write out what the abbreviations mean. If you have to use another row in the header row of tables that is OK.
Response: Thank you for this valuable suggestion. We will carefully review all tables and graphs and we will add comments underneath all tables and graphs to fully spell out the meaning of the original abbreviations.
Examples: Note: SOC: Organic carbon; TN: Total Nitrogen; TP: Total Phosphorus; TK: Total Potassium; AN: Alkaline Nitrogen; AP: Effective Phosphorus; AK: Acute Potassium; pH: Soil pH;
- There are A LOT of formatting errors and some English errors (see below). For example, one type of error that is common is the need for a space to be added. Given the potential contribution of this research to the improving our understanding of the impact of chemical fertilizers on agro-forestry yield and soil microbial diversity, this research warrants publication. Also, please make the following minor edits and clarifications:
1) MDPI Abstracts need to be no more than 200 words.
Response: Thank you for your suggestion, we have revised the abstract to reduce the text and remove redundancies.
Fertilization treatments can improve soil fertility and increase sapodilla yield, but the response of soil microbial communities in sapodilla forests to fertilization treatments and their correlation with soil fertility and sapodilla yield are not clear. For this reason, we used a combination of different levels of nitrogen (N), phosphorus (P), and potassium (K) fertilizers with and without fertilization in a field experiment to investigate the differences in soil bacterial community characteristics, soil fertility, and Sapindales yield in Sapindus indicus raw material forests under different fertilization conditions, and tried to clarify the interrelationships among the three. The results showed that (1) there were significant differences in soil properties of Sapindus indica among different fertilization treatments, and increasing the amount of N or P applied at the N-addition and P-addition levels, respectively, would significantly reduce soil pH. (2) Bacterial community diversity was significantly reduced under N3P2K2 and N1P1K2 treatments compared with no fertilization. Among the dominant soil bacterial groups at the gate level, the relative abundance of Chloroflexi showed an increasing and then decreasing trend with the increase of N application on N addition, the relative abundance of Firmicutes, Bacteroidota, and Fusobacteriota was positively correlated with P and K fertilizer application, and the relative abundance of Acidobacteriota was positively correlated with P and K fertilizer application, while that of Acidobacteriota was positively correlated with P and K fertilization. Acidobacteriota) and Verrucomicrobiota relative abundance was positively correlated with P and K fertilizer application, while Acidobacteriota and Verrucomicrobiota relative abundance decreased with increasing P and K fertilizer. (3) The highest value of saprophyte yield (1464.58 kg ha-1) was reached at N2P2K2 treatment, which increased by 258.67% compared to the control. (4) Redundancy analysis (RDA) revealed that soil pH, total K, and effective P content were the main factors affecting the bacterial community structure. (5) Structural equation modeling (SEM) analysis showed that soil nutrient content was the main direct driving factor affecting the yield of sapodilla, and the correlation between bacterial community diversity, community structure and yield was not significant. In conclusion, the rational use of formulated fertilization can change the microbial community structure and improve the microbial diversity by increasing the soil nutrient content, and at the same time improve the yield of Sapindales.
2) On L42, change to “…was obtained. The main factors”
Response: We thank the reviewer for raising this question. We've modified this part.
Redundancy analysis (RDA) revealed that soil pH, total K, and effective P content were the main factors affecting the bacterial community structure.
3) Change i.e., to e.g., since these are examples of and not another way of saying “bacterial community attributes”
Response: We thank the reviewer for raising this question. We've modified this part.
(6) Structural equation modeling showed that soil nutrient content was the main direct factor driving the yield of Sapindus mukorossi, whereas the bacterial community attributes (e.g., diversity and structure) had minor effects on the yield. In summary, the rational use of formulated fertilization can change the bacterial community structure, improve the bacterial diversity and increase the soil nutrient content, with the latter exerting a significant effect on the improvement of the yield of Sapindus mukorossi.
4) On L52-53, the keywords need to be in alphabetical order. Please make sure the first word of the first keyword is capitalized. Please change to: Keywords: Bacterial community; diversity; fertilization; Sapindus mukorossi; soil fertility; structural distribution
Response: Thanks for your suggestion. We have revised this part.
Keywords: Bacterial community; diversity; fertilization; Sapindus mukorossi; soil fertility; structural distribution
5) Throughout the manuscript, please make sure there is a space before the [#]. For example on L57, change to “…areas [1-3].
Response: Thank you. We have revised this part.
6) Also, there does no space between number for [#,#]. For example on L59, change to “…energy source [4,5].” Please make similar edits throughout the manuscript.
Response: Thank you. We have revised this part.
7) In each major section of the manuscript, define abbreviations. For example on L60, change to “…bioenergy species. Nitrogen (N), phosphorus (P), and potassium (K) are massive…”
Response: Thank you. We have revised this part.
8) Unless this is at the end of a sentence with a period, there should be a blank space on both sides of the [#]. For example on L88, change to “Ma et al. [23] showed…”
Response: Thank you. We have revised this part.
9) On L118, change to “…yield and (2) whether…”
Response: Thank you. We have revised this part.
10) In Figure 1, please increase the size of both maps so the writing can be read.
Response: Thank you. We have revised this part.
11) On L147, change to “of 2 mm to be…” where you add a space.
Response: Thank you. We have revised this part.
12) In Table 1, there does not need to be so much space between rows. Delete blank rows between rows with data.
Response: Thank you. We have revised this part.
Table 1. Combination of factors and levels in each treatment.
Processing number |
Fertilization level (kg·hm-2) |
||
N |
P |
K |
|
N0P0K0 |
0 |
0 |
0 |
N0P2K2 |
0 |
500 |
400 |
N1P2K2 |
300 |
500 |
400 |
N2P0K2 |
600 |
0 |
400 |
N2P1K2 |
600 |
250 |
400 |
N2P2K2 |
600 |
500 |
400 |
N2P3K2 |
600 |
750 |
400 |
N2P2K0 |
600 |
500 |
0 |
N2P2K1 |
600 |
500 |
200 |
N2P2K3 |
600 |
500 |
600 |
N3P2K2 |
900 |
500 |
400 |
N1P1K2 |
300 |
250 |
400 |
N1P2K1 |
300 |
500 |
200 |
N2P1K1 |
600 |
250 |
200 |
13) Write L164 as a complete sentence: “Yield measurements of Sapindus mukorossi were taken at maturity.”
Response: Thank you for your insightful comments. We fully agree that providing a more detailed description of the method is essential for the reproducibility of the experiment. We will expand on this paragraph in the Methods section.
Yield measurement: Sapindus niloticus was harvested at fruit maturity (November 5, 2021), and waited until all the fruits were ripe for picking the fruits of the whole plant, while Sapindus niloticus single-plant yield refers to the fruit yield of each plot by picking all the fruits that have matured from the whole tree, and then weighed the fruits using an electronic scale after picking, and then counted.
14) Add L164-166 to the end of the paragraph on L163.
Response: Thanks for your suggestion. We have revised this part.
15) All weblinks on L183-191 need to be put in [#] format and cited as references. See MDPI Microorganisms Word template for authors for citation formatting for this (accessed 30 December 2024).
Response: Thank you. We have revised this part.
16) On L211, change to 3. Results
Response: Thanks for your suggestion. We have revised this part.
17) Everywhere in the manuscript, use the sub-script formatting. There will be a lot of edits to make. For example: N2P2K2
Response: Thank you. We have revised this part.
18) For Table 2, please widen columns and increase size of font for whole table.
Response: Thank you. We have revised this part.
19) For both Figure 2 and Figure 3, put one graph on top of the other and increase size of both graphs so the writing can be read.
Response: Thank you for your insightful comments. We have revised this part.
20) Change to Figure 3. On L289 and do not abbreviate.
Response: Thank you. We have revised this part.
21) For Table 4, widen the first column so all words fit on one row.
Response: Thank you. We have revised this part.
Table 4. Relationship between bacterial community diversity, fruit yield and soil environmental factors.
|
pH |
TN |
TP |
TK |
SOC |
AN |
AP |
AK |
OTU richness |
0.336* |
0.110 |
-0.345* |
-0.097 |
-0.211 |
-0.047 |
-0.109 |
0.288 |
Simpson |
0.542** |
-0.062 |
-0.532** |
-0.248 |
-0.200 |
-0.172 |
-0.349* |
0.126 |
Shannon |
0.484** |
0.044 |
-0.445** |
-0.149 |
-0.185 |
-0.036 |
-0.275 |
0.331* |
yield |
-0.699** |
0.645** |
0.634** |
0.405 |
0.699** |
0.773** |
0.707** |
0.378 |
Note: * indicates significant correlation at the 0.05 level; ** indicates significant correlation at the 0.01 level.
22) Here and everywhere else, write out Fig. as Figure such as on L325 where this should be (Figure 5)
Response: Thank you. We have revised this part.
23) The Note on L353-361 should be moved to the writing.
Response: Thank you. We have revised this part.
24) In each major section of the manuscript, define abbreviations. For example on
L530, change to “…combinations of nitrogen (N), phosphorus (P), and potassium (K)
can significantly change…”
Response: Thanks. We have revised this part.
25) In the References, please put journal names in italics so for example Forest Ecol.
Manag. on L560. Please also add DOI links at the end of each reference.
Response: Thanks. We have revised this part.

Reviewer 2 Report
Comments and Suggestions for Authors
Review report
The study explores the interplay between fertilization, soil microbial communities, and Sapindus mukorossi yield. The findings provide valuable insights into how fertilization affects soil health and productivity. While the research addresses an important topic, some aspects of methodology, data interpretation, and presentation could be refined to enhance clarity and impact.
Comments for authors
1. The sampling depth is stated as 0–20 cm, but no justification is provided for this depth, particularly for a forest soil where deeper layers might contribute to nutrient cycling and microbial activity.
2. The "five-point sampling method" is described without elaboration on how sampling points were determined (e.g., were they equidistant or randomly distributed?).
3. There is no mention of controls for sequencing accuracy, such as mock communities or blank extractions to check for contamination.
4. The one-way ANOVA approach does not account for potential interactions between N, P, and K applications, which could influence soil properties and microbial communities. A factorial ANOVA or mixed-effects model might provide more robust insights.
5. The statement "Refer to the literature[8,9], for specific details of soil nutrient indicators" is inadequate; critical information should be included directly in the manuscript.
6. Details about how the soil samples were processed (e.g., sieving, homogenization) for microbial and chemical analyses are provided but lack specific information on protocols to minimize contamination or variability.
7. No mention is made of whether the experiment was repeated over multiple years or growing seasons to account for interannual variability in soil and plant responses.
By addressing these shortcomings, the materials and methods section will better support the study's findings and enhance the reproducibility and reliability of the research.
Author Response
Dear Editor-in-Chief:
We have revised our previously-submitted manuscript entitled “Fertilization Induced Soil Microbial Shifts Show Minor Effects on Sapindus mukorossi Yield” (Manuscript ID: 3400875) by carefully following the comments by the three reviewers and the editor. Now, we resubmit the new manuscript based on the suggestions.
In this revised version we have responded to each of the issues raised by the reviewers to improve both the scope and expression of the work. The major changes in the current revision include:
- We have revised the introductory section as suggested by reviewer #1 to introduce the importance of background information and economic value of Sapindus spp. We have also revised the Materials and Methods section as suggested by the reviewer by revising the whole text such as blank lines and fonts, deleting the redundant text, and refining the content to make it clearer.
- We have explained the problems as suggested by reviewer #2 “why the sampling depth was 0-20 cm” and “no mention of whether the experiment was repeated over multiple years or growing seasons”.
- We have followed the recommendations of reviewer 3 on “Rationale for the choice of fertilizer trial and fertilizer design” and “Frequency of yield measurements and whether inter-annual variability was taken into account” and “Reasons for optimization of yield by N2P2K2 treatment”. and “Reasons for optimizing yield with N2P2K2 treatment” were explained.
The detailed responses to specific comments are listed below.
Yours Sincerely,
Juntao Liu
Ph.D.
Reviewer #2:
The study explores the interplay between fertilization, soil microbial communities, and Sapindus mukorossi yield. The findings provide valuable insights into how fertilization affects soil health and productivity. While the research addresses an important topic, some aspects of methodology, data interpretation, and presentation could be refined to enhance clarity and impact.
- The sampling depth is stated as 0–20 cm, but no justification is provided for this depth, particularly for a forest soil where deeper layers might contribute to nutrient cycling and microbial activity.
Response: Thanks for your suggestion. Our study focuses on nutrient cycling and microbial activity in the top soil layer (0-20 cm) because this layer is the area where plant roots are mainly distributed and has a direct impact on plant growth and ecosystem function. The physicochemical properties and biological characteristics of the top soil layer are highly variable, so the study of this layer is of great importance. In the relevant literature we reviewed, many studies on forest soils also used a sampling depth of 0-20 cm. These studies show that the top soil layer has the most significant impact on the ecosystem. Although we recognize the potential role of deeper soils in nutrient cycling and microbial activity, based on our preexperimental results and literature review, the 0-20 cm soil layer is already a better reflection of the study area as a whole. Future studies may consider expanding the sampling depth to gain a more comprehensive understanding of soil properties.
- The "five-point sampling method" is described without elaboration on how sampling points were determined (e.g., were they equidistant or randomly distributed?).
Response: Thank you for your careful review and correction of the inadequacy of the description of the “five-point sampling method”. First, we identified specific sampling areas based on the basic conditions of the experimental plots and the regional characteristics of each fertilization area. These areas were selected to be representative of typical soil characteristics within the sapodilla woodland under each fertilization treatment. Sampling points were arranged within each sampling area following a systematic sampling method. Specifically, sampling points were evenly distributed at regular intervals (equidistant) within the area. In order to reduce errors due to local variations, the soil under each fertilizer treatment is then mixed samples can better cover the soil nutrient and microbial bacterial communities of the soil samples from the plots of each fertilizer treatment, and this method ensures that the samples are representative of the soil characteristics of the whole area. We will also randomly select specific sampling locations within a certain range based on the intended proximity of the sampling points in each fertilized plot to avoid the effects of possible local variability.
- There is no mention of controls for sequencing accuracy, such as mock communities or blank extractions to check for contamination.
Response: Thank you. Thank you for your valuable comments. Regarding the control measures for sequencing accuracy, such as the use of mock communities or blank extractions, in our study, since this trial is a field experiment, which is a test of soil properties and microbial communities under different fertilization treatments, due to the specificity of the trial design, the company that we sent the test to for testing did not introduce any contamination in the amplification process, but it did not amplify. Did a negative control can prove that the amplification link did not introduce pollution, but it did not amplify the subsequent we will adopt potting control test using simulated communities or blank extraction as a control of sequencing accuracy. In the field trial, our main focus was on the effect of fertilization treatments on soil microbial community structure. Nonetheless, we understand the importance of control experiments to validate the accuracy of the experimental process and to detect potential contamination. These blank extract samples were processed alongside the actual samples throughout the experimental process to detect and rule out possible contamination, but it was not amplified. Use strict laboratory protocols to minimize the risk of cross-contamination. Strict quality control of sequencing data, including removal of low-quality reads and possible chimeric sequences. Use standard operating procedures for high-throughput sequencing platforms that have been validated by the platform provider. We will include this information in the revised manuscript and thank you for helping us improve the quality of the paper.
- The one-way ANOVA approach does not account for potential interactions between N, P, and K applications, which could influence soil properties and microbial communities. A factorial ANOVA or mixed-effects model might provide more robust insights.
Response: Thank you for your careful consideration of our statistical analysis methods and for your suggestions. We agree with you about the importance of considering potential interactions between nitrogen (N), phosphorus (P), and potassium (K) fertilization. In the current manuscript, we have used a variety of statistical methods such as redundancy analysis (RDA), correlation analysis, and structural equation modeling (SEM) to explore the effects of fertilization treatments on soil properties and microbial communities. These methods have indeed provided us with insights into treatment effects and relationships between variables. Due to the specificity of our fertilization treatments, which can be at the nitrogen level (N0P2K2, N1P2K2, N2P2K2, N3P2K2), phosphorus level (N2P0K2, N2P1K2, N2P2K2, N2P3K2), and potassium level (N2P2K0, N2P2K1, N2P2K2, N2P2K3), and given the fact that the current manuscript has been lengthy and richer, we focused on presenting the main results and findings at the time of writing. To keep the article focused and readable, we did not include a detailed analysis of the interaction between N, P, and K fertilization. In the future, we will explore in detail the interactions among N, P, and K fertilization in our next article, which will be devoted to an in-depth analysis of the effects of these interactions on soil ecosystems in sapodilla woodlands. We believe that such an arrangement will be able to communicate our findings more effectively and provide readers with a more comprehensive understanding.
- The statement "Refer to the literature[8,9], for specific details of soil nutrient indicators" is inadequate; critical information should be included directly in the manuscript.
Response: Thanks for your suggestion. We understand your concern regarding the inadequacy of the statement “Please refer to the literature for specific details on soil nutrient indices [8,9]”, and we agree that information that is essential to the understanding of the results of this study should be included directly in the manuscript to ensure that readers do not need to rely on the literature to fully understand the content. We are adding a short paragraph to the revised manuscript, and we will still retain references to previous studies, but will ensure that the information in the current manuscript is sufficiently complete.
The study site was located in Jianning County, Sanming City, Fujian Province, China (116°47′20″E, 26°40′3″N), with an average annual temperature of 17.0°C, an average annual rainfall of 1792 mm, and a relative humidity of 84%. The soil at the experimental site was a sandy clay loam, with a soil organic carbon content of 7.75 g.kg-1, a total nitrogen content of 1.40 g.kg-1, a total phosphorus content of 0.36 g.kg-1, and a total potassium content of 27.95 g.kg-1. The total potassium content was 27.95 g.kg-1, quick-acting potassium content was 48.16 mg.kg-1, effective phosphorus content was 1.32 mg.kg-1, and alkaline dissolved nitrogen content was 36.01 mg.kg-1. The raw material forest of Sapindus saprophyllus was the asexual line of 'Yuanhua', with an average height of 2.38 m, an average diameter of 6.97 cm, and an average crown width of 2.4 m × 1.4 m. The average height was 2.38 m, with an average diameter of 6.97 cm. The average tree height was 2.38m, the average diameter was 6.97cm, and the average crown width was 2.4m×2.2m.
- Details about how the soil samples were processed (e.g., sieving, homogenization) for microbial and chemical analyses are provided but lack specific information on protocols to minimize contamination or variability.
Response: Thank you for your interest in the details of soil sample handling. We recognize that the description of soil sample handling (e.g., sieving, homogenization) in the original manuscript needs further refinement, especially in terms of reducing contamination and variability. To ensure the rigor of the study and the credibility of the results, we will make additions to the revised manuscript in this regard. Describe in detail the aseptic handling techniques employed during sampling, sieving, and homogenization, including the use of sterile tools, gloves, and sterilization procedures for the laboratory environment. Specify sieve size for sieving and specific methods of homogenization to ensure consistency and representativeness of samples. Use of separate laboratory space, equipment cleaning and sterilization procedures before and after sample processing.
- No mention is made of whether the experiment was repeated over multiple years or growing seasons to account for interannual variability in soil and plant responses.
Response: Thank you for your review and suggestions. Regarding the question of whether the experiment was repeated in multiple years or growing seasons to account for interannual variability in soil and plant responses, in our study this experiment was indeed conducted in a single year to explore patterns of change in soil properties and soil microbial-bacterial communities following fertilizer application. We recognize that this design may not adequately account for the interannual variability in soil and plant responses, which is an important point you make. We fully agree with you on the importance of experiments across years to assess long-term soil and plant responses. We will direct our future research toward experiments that span years to verify the stability and generalizability of results.

Reviewer 3 Report
Comments and Suggestions for Authors
This manuscript provides valuable insights into how different fertilization regimes affect soil microbial communities and the yield of Sapindus mukorossi, a key bioenergy species.
Several aspects require further refinement to enhance clarity, depth, and rigor_
Lines 16–51: Expand on the novelty of the study by emphasizing how these findings advance current knowledge. Quantify key results (e.g., percentage increase in yield or shifts in microbial diversity).
Lines 55–119:
Provide more context about the ecological and economic importance of Sapindus mukorossi. Discuss specific challenges in its cultivation that this study addresses.
Clarify the gaps in knowledge regarding the interactions between soil nutrients, microbial communities, and yield outcomes.
Lines 121–150: Include justification for the chosen fertilization levels and experimental design. Were they based on prior studies or preliminary experiments?
Specify how the "five-point sampling method" was adapted for microbial analyses and whether spatial variability within plots was accounted for.
Lines 154–192: Provide more details about DNA extraction and sequencing protocols, including quality control steps and sequencing depth.
Clarify statistical methods used to analyze microbial diversity indices and redundancy analysis (RDA).
Lines 164–166: Discuss the frequency of yield measurement and whether inter-annual variability was considered.
Lines 212–231: Include a more detailed discussion on why some nutrient levels showed a non-linear response to increasing fertilizer inputs.
Highlight key differences in soil pH and organic carbon trends across treatments.
Microbial Community Structure:
Lines 232–263: Expand on the ecological implications of observed shifts in dominant bacterial phyla. For instance, how might increases in Proteobacteria or decreases in Acidobacteriota affect nutrient cycling?
Relate findings from principal component analysis (PCA) to functional roles of microbial taxa.
Lines 266–285: Provide more context for why the N2P2K2 treatment optimized yield. Were specific microbial or soil chemical parameters uniquely associated with this treatment?
Lines 362–531: Deepen the discussion on how shifts in microbial communities mediate soil nutrient dynamics and plant productivity. Link findings to broader ecological theories or models.
Address potential trade-offs, such as soil acidification under high N or P treatments, and suggest mitigation strategies.
Comparative Analysis:
Compare results with similar studies in other bioenergy crops or plantation systems. Highlight unique findings for Sapindus mukorossi.
Propose follow-up studies, such as testing fertilization strategies under different climatic or soil conditions.
Annotate key trends in PCA or RDA plots to guide interpretation.
Include additional information, such as variance explained by diversity indices, in Table 3.
Lines 523–540: Strengthen the conclusion by emphasizing the broader implications of the study for sustainable agriculture and soil health management.
Author Response
Dear Editor-in-Chief:
We have revised our previously-submitted manuscript entitled “Fertilization Induced Soil Microbial Shifts Show Minor Effects on Sapindus mukorossi Yield” (Manuscript ID: 3400875) by carefully following the comments by the three reviewers and the editor. Now, we resubmit the new manuscript based on the suggestions.
In this revised version we have responded to each of the issues raised by the reviewers to improve both the scope and expression of the work. The major changes in the current revision include:
- We have revised the introductory section as suggested by reviewer #1 to introduce the importance of background information and economic value of Sapindus spp. We have also revised the Materials and Methods section as suggested by the reviewer by revising the whole text such as blank lines and fonts, deleting the redundant text, and refining the content to make it clearer.
- We have explained the problems as suggested by reviewer #2 “why the sampling depth was 0-20 cm” and “no mention of whether the experiment was repeated over multiple years or growing seasons”.
- We have followed the recommendations of reviewer 3 on “Rationale for the choice of fertilizer trial and fertilizer design” and “Frequency of yield measurements and whether inter-annual variability was taken into account” and “Reasons for optimization of yield by N2P2K2 treatment”. and “Reasons for optimizing yield with N2P2K2 treatment” were explained.
The detailed responses to specific comments are listed below.
Yours Sincerely,
Juntao Liu
Ph.D.
Reviewer #3:
This manuscript provides valuable insights into how different fertilization regimes affect soil microbial communities and the yield of Sapindus mukorossi, a key bioenergy species.
Response: Thank you for your recognition to our work. We have responded to your comments very carefully. The detailed responses to specific comments are listed below.
- Lines 16–51: Expand on the novelty of the study by emphasizing how these findings advance current knowledge. Quantify key results (e.g., percentage increase in yield or shifts in microbial diversity).
Response: Thank you for your suggestions and interest in our research. In response to your suggestions about emphasizing the novelty of the study and quantifying the key findings. We have revised the abstract section to more clearly highlight the innovations and key findings of this study. The abstract now includes quantitative data on yield increases, as well as specific descriptions of changes in microbial diversity.
Fertilization treatments can improve soil fertility and increase sapodilla yield, but the response of soil microbial communities in sapodilla forests to fertilization treatments and their correlation with soil fertility and sapodilla yield are not clear. For this reason, we used a combination of different levels of nitrogen (N), phosphorus (P), and potassium (K) fertilizers with and without fertilization in a field experiment to investigate the differences in soil bacterial community characteristics, soil fertility, and Sapindales yield in Sapindus indicus raw material forests under different fertilization conditions, and tried to clarify the interrelationships among the three. The results showed that (1) there were significant differences in soil properties of Sapindus indica among different fertilization treatments, and increasing the amount of N or P applied at the N-addition and P-addition levels, respectively, would significantly reduce soil pH. (2) Bacterial community diversity was significantly reduced under N3P2K2 and N1P1K2 treatments compared with no fertilization. Among the dominant soil bacterial groups at the gate level, the relative abundance of Chloroflexi showed an increasing and then decreasing trend with the increase of N application on N addition, the relative abundance of Firmicutes, Bacteroidota, and Fusobacteriota was positively correlated with P and K fertilizer application, and the relative abundance of Acidobacteriota was positively correlated with P and K fertilizer application, while that of Acidobacteriota was positively correlated with P and K fertilization. Acidobacteriota) and Verrucomicrobiota relative abundance was positively correlated with P and K fertilizer application, while Acidobacteriota and Verrucomicrobiota relative abundance decreased with increasing P and K fertilizer. (3) The highest value of saprophyte yield (1464.58 kg ha-1) was reached at N2P2K2 treatment, which increased by 258.67% compared to the control. (4) Redundancy analysis (RDA) revealed that soil pH, total K, and effective P content were the main factors affecting the bacterial community structure. (5) Structural equation modeling (SEM) analysis showed that soil nutrient content was the main direct driving factor affecting the yield of sapodilla, and the correlation between bacterial community diversity, community structure and yield was not significant. In conclusion, the rational use of formulated fertilization can change the microbial community structure and improve the microbial diversity by increasing the soil nutrient content, and at the same time improve the yield of Sapindales.
- Lines 55–119:Provide more context about the ecological and economic importance of Sapindus mukorossi. Discuss specific challenges in its cultivation that this study addresses.Clarify the gaps in knowledge regarding the interactions between soil nutrients, microbial communities, and yield outcomes.
Response: Thank you for your valuable suggestions, which were very helpful in enhancing the depth and breadth of our study. We will add a paragraph in the introduction section to discuss in detail the importance of the ecological and economic values of Sapindus mukorossi, such as its applications in the pharmaceutical, cosmetic, and cleaning product industries, as well as its potential economic benefits and market prospects. We also dedicate the third paragraph of the introduction to the knowledge gap on the interactions between soil nutrients, microbial communities and yield outcomes.
Sapindus mukorossi is an energy forest tree species of the genus Sapindus in the Sapindaceae family, which is mainly distributed in tropical and subtropical regions (Xu et al. 2021; Liu et al. 2022b; Gao et al. 2018). With the increasing global demand for energy, bioenergy species have received widespread attention as an emerging and promising energy resource (Xu et al. 2023). Sapindus mukorossi is widely used in biomedicine, bioenergy, and cosmetics (Liu et al. 2022; Wang et al. 2021; Xu et al. 2021; Zhao et al. 2019). The whole plant of Sapindus indica, including its fruits, roots, bark and leaves, has been used in traditional medicine in China (Xu et al. 2023). Its soap pods, with a crude protein content of about 22%, and the kernels of the soap berries, which contain about 40% fatty acids, are also woody oilseed species that have been promoted for use in recent years (Xu et al. 2022). Only in Fujian Province, which is one of its main production areas, the planting area reaches 20,000 ha. However, the yield and quality of bioenergy species are mainly affected by soil nutrient conditions, and N, P, and K are a large number of nutrients necessary for plant growth, which play a crucial role in crop yield and oil product synthesis, and the rational application of N, P, and K fertilizers is an effective strategy to achieve high yield and high quality of economic tree species (Sun et al. 2022; Li et al. 2019b). However, in actual production, farmers usually over-apply N, P, and K fertilizers in order to rapidly increase the fruit yield of sapodilla. Unscientific fertilization measures can cause into problems such as soil quality decline, acidification and sloughing, which ultimately lead to crop yield reduction, as well as serious environmental problems such as groundwater source pollution (Xue et al. 2020; Luo et al. 2016). Exploring scientific fertilization ratios is essential to enhance the sustainable development of the Sapindus industrial forest.
- Lines 121–150: Include justification for the chosen fertilization levels and experimental design. Were they based on prior studies or preliminary experiments?Specify how the "five-point sampling method" was adapted for microbial analyses and whether spatial variability within plots was accounted for.
Response: Thank you for your careful review and suggestions. We will add a paragraph to the revised manuscript detailing the basis for the selection of fertilizer application levels. These fertilizer application amounts and levels are based on the results of our team's previous research and the preliminary experiments we conducted, combined with the soil background data of our experimental area to arrive at the fertilizer application amounts for this experiment. First, we identified specific sampling areas based on the basic conditions of the experimental sample plots and the regional characteristics of each fertilizer application area. These areas were selected to be representative of typical areas of soil characteristics within the sapodilla woodland under each fertilizer application treatment. Sampling points were arranged within each sampling area following a systematic sampling method. Specifically, sampling points were evenly distributed at regular intervals within the area. In order to reduce errors due to local variations, the soil under each fertilizer treatment is then mixed samples can better cover the soil nutrients and microbial bacterial communities of the soil samples from the plots of each fertilizer treatment, and this method ensures that the samples are representative of the entire region. We will also randomly select specific sampling locations within a certain range based on the intended proximity of the sampling points in each fertilized plot to avoid the effects of possible local variability.
- Lines 154–192: Provide more details about DNA extraction and sequencing protocols, including quality control steps and sequencing depth.Clarify statistical methods used to analyze microbial diversity indices and redundancy analysis (RDA).
Response: Thank you for your careful review and suggestions in the Methods section of our study. We will describe the DNA extraction process in detail in the revised manuscript, including the kits used, extraction steps, and any specific optimization measures. For sequencing protocols, we will provide the complete sequencing process, including library construction, sequencing platform, sequencing strategy, and primers and probes used. We will also detail the quality control steps, including assessment of DNA concentration and purity, control of library quality, and determination of sequencing depth. We will provide specific procedures and standards to ensure data reliability and reproducibility. We will clearly describe the statistical methods used to analyze microbial diversity indices. This includes the software used, the formulas for calculations, and how they were handled. SPSS 21.0 one-way analysis of variance (ANOVA) and least significant difference multiple range test (P Ë‚ 0. 05) were utilized in order to assess the effect of different NPK fertilizers on soil properties (TN, TP, TK, SOC, AN, AP, AK, pH), soil bacterial community diversity indices (OTU richness, Simpson, Shannon indices) , and soil microbial community composition. Redundancy analysis was used to test the relationship between soil microbial bacterial community and soil properties under different fertilization treatments and was mapped using Canoco 5 software.
- Lines 164–166: Discuss the frequency of yield measurement and whether inter-annual variability was considered.
Response: Thank you for your review. In our study, this experiment was indeed conducted only within a single year to explore the changing patterns of soil properties and soil microbial bacterial communities after fertilizer application. Since Sapindus indica is a woody oilseed tree species, the fruit matures in early November every year, and the leaves spread from April to December, this experiment is to explore the changes of soil bacterial community characteristics in Sapindus indica woodland after fertilization, analyze the community characteristics and changing rules of soil bacteria in response to nitrogen, phosphorus and potassium fertilizers, screen for key soil nutrient factors affecting soil bacterial community, explore the effects of fertilization formulations on Sapindus indica soil microbial mechanism, and to reveal the interrelationships among soil bacterial community characteristics, soil fertility and Sapindus indicus yield under fertilization treatments. We recognize that this design may not adequately account for the interannual variability of soil and plant responses, an important point you make. We fully agree with you about the importance of experiments across years to assess long-term soil and plant responses. We will direct our future research toward experiments that span years to verify the stability and generalizability of results.
- Lines 212–231: Include a more detailed discussion on why some nutrient levels showed a non-linear response to increasing fertilizer inputs.Highlight key differences in soil pH and organic carbon trends across treatments.
Response: We thank the reviewer for raising this question. We have added a detailed analysis of differences in soil pH and organic carbon trends to the discussion section.
In this study, it was found that the soil total and available nitrogen contents showed an increase and then a decrease trend with the increase of nitrogen application, which was inconsistent with the results of Qiao et al.[36], and could be explained by the theory of "stoichiometric decomposition" proposed by Hessen et al. [38]. The stoichiometric imbalance after nitrogen addition has been shown to accelerate soil organic carbon decomposition to maintain soil carbon: nitrogen ratios, with decomposition rates and microbial activity being highest when carbon and nitrogen inputs corresponded to microbial stoichiometric carbon to nitrogen ratios[39, 40]. In a previous study conducting in energy forests, N additions accompanied meet the needs of microbial growth and stimulate the mineralization of native soil organic matter as well as the release of reactive N during nitrification, denitrification, and leaching[31] . This highlights the importance of rational application of nitrogen fertilizers.
- Lines 232–263: Expand on the ecological implications of observed shifts in dominant bacterial phyla. For instance, how might increases in Proteobacteria or decreases in Acidobacteriota affect nutrient cycling?Relate findings from principal component analysis (PCA) to functional roles of microbial taxa.
Response: Thanks for your suggestion. We have revised this part.
- Lines 266–285: Provide more context for why the N2P2K2 treatment optimized yield. Were specific microbial or soil chemical parameters uniquely associated with this treatment?
Response: Thanks for your suggestion. We have revised this part.
In this study, it was found that N2P2K2 treatment was the best fertilizer application rate for the composite index and over fertilization reduced the yield. When the fertilizer application did not reach the threshold, with the application of nitrogen, phosphorus and potassium yield increase may be due to the increase in soil nutrients with the increase in fertilizer application. This is shown by the increase in both soil effective phosphorus and organic carbon content. In addition, another reason contributing to the increase in yield may be the increased resistance of the tree itself caused by the application of fertilizer, the reduction of diseases in the plant, the reduction of insect pests, and other factors. However, when the rate of fertilizer application exceeds the threshold, increasing the amount of fertilizer application can have a negative effect on the plant. The reason for this negative effect may be (1) when the fertilizer application exceeds the threshold, the nutrient supply capacity inside the soil is reduced instead, and the ability to contribute to the yield is weakened, resulting in lower yields. Another reason is that the higher content of leaves when they are high in nitrogen may imply leaf senescence and delayed growth of other nutrient organs, which leads to consumption of more carbon skeleton and energy by the fruit or storage organs and low accumulation of carbohydrates. (2) Excessive fertilizers can be harmful to the environment. Relevant studies have shown that fertilizer is one of the main sources of soil, water and air pollution. Excessive fertilizers can lead to serious soil sloughing, resulting in the imbalance of soil nutrient structure, which in turn restricts the growth and development of the plant root system, reduces the ability to absorb nutrients and water, and ultimately leads to lower yields. (3) In addition, over-fertilization will also increase plant diseases. For example, studies on diseases of Panax quinquefolium, maize and citrus showed that over-fertilization increased the incidence of root rot in Panax quinquefolium and brown spot in coffee as well as induced severe defoliation, resulting in adverse short- and long-term consequences. The present study also found that many new branches sprouted on the whole tree of Sapindus indica in high fertilization and the incidence of coal stain disease on leaves increased. Therefore, rational application of nitrogen, phosphorus and potassium fertilizers will be effective in improving soil and leaf physiological attributes in sapodilla stands, thus playing an important role in increasing sapodilla yield.
- Lines 362–531: Deepen the discussion on how shifts in microbial communities mediate soil nutrient dynamics and plant productivity. Link findings to broader ecological theories or models. Address potential trade-offs, such as soil acidification under high N or P treatments, and suggest mitigation strategies.
Response: Thank you very much for your in-depth review of our paper and your valuable suggestions for the discussion section. We recognize that while the discussion touches on the relationship of microbial community change to soil nutrient dynamics and plant productivity, we will expand on this discussion by elaborating on how microbial community change regulates plant productivity by affecting soil nutrient transformation, uptake, and cycling. We will explore the implications of these trade-offs for soil health and ecosystem services and make practical recommendations based on our findings, such as adapting fertilization strategies, using soil amendments, or adopting crop rotation systems.
10.Compare results with similar studies in other bioenergy crops or plantation systems. Highlight unique findings for Sapindus mukorossi.Propose follow-up studies, such as testing fertilization strategies under different climatic or soil conditions.Annotate key trends in PCA or RDA plots to guide interpretation.Include additional information, such as variance explained by diversity indices, in Table 3.Lines 523–540: Strengthen the conclusion by emphasizing the broader implications of the study for sustainable agriculture and soil health management.
Response: Thank you for your your suggestions of what we have added to the manuscript. Also the broader implications of this research for sustainable agriculture and soil health management.

Reviewer 4 Report
Comments and Suggestions for Authors
In the article presented field experiment concerning the influence of fertilization using nitrogen, phosphorous and potasium on soil nutrients and bacterial communities for correlation with the yield of Sapindus mukorossi. Added nitrogen and phosphorous provide the increase of degree of soil acidification, what can led to decomposition of organic carbon in the soil. Presented experiment showed the increase of soil productivity and made lower environmental pollution when using fertilizer application strategies. Besides that the substantial impact on the structure of soil bacterial communities was obtained.
Part of the Section Abstract should be moved to Section Introduction and part of it should be moved to Section Discussion.
In Section Coclusions should be added Sub Sections with a description of the most important findigs of the study.
Author Response
Dear Editor-in-Chief:
We have revised our previously-submitted manuscript entitled “Fertilization Induced Soil Microbial Shifts Show Minor Effects on Sapindus mukorossi Yield” (Manuscript ID: 3400875) by carefully following the comments by the two reviewers and the editor. Now, we resubmit the new manuscript based on the suggestions.
In this revised version we have responded to each of the issues raised by the reviewers to improve both the scope and expression of the work. The major changes in the current revision include:
(1)We have added calculation formulas and reorganized the sentence structure of the article as suggested by Reviewer 1. We have also made revisions to the Materials and Methods section as suggested by Reviewer 1, including modifying the entire text such as blank lines and fonts, removing unnecessary text, and refining the content to make it clearer.
(2)We have revised the abstract and discussion sections according to Reviewer # 2's questions: 'Some chapter abstracts should be moved to chapter introductions, and some chapters should be moved to chapter discussions'. Simplified abstract text and refined article content
The detailed responses to specific comments are listed below.
Yours Sincerely,
Juntao Liu
Ph.D.
Reviewer #2:
In the article presented field experiment concerning the influence of fertilization using nitrogen, phosphorous and potasium on soil nutrients and bacterial communities for correlation with the yield of Sapindus mukorossi. Added nitrogen and phosphorous provide the increase of degree of soil acidification, what can led to decomposition of organic carbon in the soil. Presented experiment showed the increase of soil productivity and made lower environmental pollution when using fertilizer application strategies. Besides that the substantial impact on the structure of soil bacterial communities was obtained.
Response: Thank you for your recognition to our work. We have responded to your comments very carefully. The detailed responses to specific comments are listed below.
1.Part of the Section Abstract should be moved to Section Introduction and part of it should be moved to Section Discussion.
Response: Thank you for your suggestion, we have revised the abstract to reduce the text and remove redundancies.
Fertilization treatments can improve soil fertility and increase sapodilla yield, but the response of soil microbial communities in sapodilla forests to fertilization treatments and their correlation with soil fertility and sapodilla yield are not clear. For this reason, we used a combination of different levels of nitrogen (N), phosphorus (P), and potassium (K) fertilizers with and without fertilization in a field experiment to investigate the differences in soil bacterial community characteristics, soil fertility, and Sapindales yield in Sapindus indicus raw material forests under different fertilization conditions, and tried to clarify the interrelationships among the three. The results showed that (1) there were significant differences in soil properties of Sapindus indica among different fertilization treatments, and increasing the amount of N or P applied at the N-addition and P-addition levels, respectively, would significantly reduce soil pH. (2) Bacterial community diversity was significantly reduced under N3P2K2 and N1P1K2 treatments compared with no fertilization. Among the dominant soil bacterial groups at the gate level, the relative abundance of Chloroflexi showed an increasing and then decreasing trend with the increase of N application on N addition, the relative abundance of Firmicutes, Bacteroidota, and Fusobacteriota was positively correlated with P and K fertilizer application, and the relative abundance of Acidobacteriota was positively correlated with P and K fertilizer application, while that of Acidobacteriota was positively correlated with P and K fertilization. Acidobacteriota) and Verrucomicrobiota relative abundance was positively correlated with P and K fertilizer application, while Acidobacteriota and Verrucomicrobiota relative abundance decreased with increasing P and K fertilizer. (3) The highest value of saprophyte yield (1464.58 kg ha-1) was reached at N2P2K2 treatment, which increased by 258.67% compared to the control. (4) Redundancy analysis (RDA) revealed that soil pH, total K, and effective P content were the main factors affecting the bacterial community structure. (5) Structural equation modeling (SEM) analysis showed that soil nutrient content was the main direct driving factor affecting the yield of sapodilla, and the correlation between bacterial community diversity, community structure and yield was not significant. In conclusion, the rational use of formulated fertilization can change the microbial community structure and improve the microbial diversity by increasing the soil nutrient content, and at the same time improve the yield of Sapindales.
2.In Section Coclusions should be added Sub Sections with a description of the most important findigs of the study.
Response: Thank you for your meticulous review and valuable feedback on our research manuscript. We fully agree with your suggestion to add a subsection in the conclusion section to describe the most important findings of the study. But most English articles are written in this way, because the abstract clearly indicates results 1, 2, 3, 4, and 5. If the conclusion is in this form, it overlaps with the abstract. I suggest using the original text content.

Round 2
Reviewer 1 Report
Comments and Suggestions for Authors
Microorganisms
Manuscript Draft
Manuscript Number: 3400857
Title: Fertilization Induced Soil Microbial Shifts Show Minor Effects on Sapindus mukorossi Yield
Article Type: Research article
Please make the following edits to the manuscript:
1) In the Methods section, please add equations and writing describing the calculations of a) OTU richness, b) Simpson diversity index, and c) Shannon diversity index. This was not done. For example, they should look like:
a) Define OTU richness verbally in the writing
b) Simpson diversity index: (1)
c) Shannon diversity index: (2)
Please use above MDPI formatting for equations including (#) fully right justified.
2) Edit L188-192:
Yield measurements for Sapindus mukorossi were made during harvest at fruit maturity on 5 November 2021. The fruit was all ripe when the plants were picked. For fruit yield for each plot of Sapindus mukorossi involved picking all the ripe fruits from each whole tree, weighing the fruits using an electronic scale after picking, and then summing the fruit yield per tree for each plot.
Recall that a solid paragraph is at least 3 sentences (1 topic sentence followed by a minimum of two supporting sentences).
3) On L87, delete space so change to: [11,14]
4) Delete blank row on L195
5) On L201, delete space so change to: [2,28]
6) On L342 this should be: Figure 3. Yield and…
7) For ALL Tables, increase font size to Palatino Linotype size 10
8) Please check the English of all writing that was recently added to the manuscript. For example, the sentence added on L593-595 reads:
Meanwhile the broader implications of this research for sustainable agriculture and soil health management.
This is NOT a complete sentence since it lacks a verb.
9) In the References, please put journal names in italics so for example Forest Ecol. Manag. on L560. This edit was not done.
10) Author contribution on L583-595 lack all the specific roles and responsibilities from the original template. Please add these back in and specify which co-authors did what.
Comments on the Quality of English LanguageThe English is needs to be improved for the writing that was added.
Author Response
Dear Editor-in-Chief:
We have revised our previously-submitted manuscript entitled “Fertilization Induced Soil Microbial Shifts Show Minor Effects on Sapindus mukorossi Yield” (Manuscript ID: 3400875) by carefully following the comments by the two reviewers and the editor. Now, we resubmit the new manuscript based on the suggestions.
In this revised version we have responded to each of the issues raised by the reviewers to improve both the scope and expression of the work. The major changes in the current revision include:
(1)We have added calculation formulas and reorganized the sentence structure of the article as suggested by Reviewer 1. We have also made revisions to the Materials and Methods section as suggested by Reviewer 1, including modifying the entire text such as blank lines and fonts, removing unnecessary text, and refining the content to make it clearer.
(2)We have revised the abstract and discussion sections according to Reviewer # 2's questions: 'Some chapter abstracts should be moved to chapter introductions, and some chapters should be moved to chapter discussions'. Simplified abstract text and refined article content
The detailed responses to specific comments are listed below.
Yours Sincerely,
Juntao Liu
Ph.D.
Reviewer #1:
This manuscript is written clearly, is well-structured, and is potentially relevant to the field since it focuses on the yield impacts of chemical fertilizers on crop/soil microbes. However, the manuscript needs to make the following FIVE general clarifications:
Response: Thank you for your recognition to our work. We have responded to your comments very carefully. The detailed responses to specific comments are listed below.
- In the Methods section, please add equations and writing describing the calculations of a) OTU richness, b) Simpson diversity index, and c) Shannon diversity index. This was not done. For example, they should look like:
- a) Define OTU richness verbally in the writing
- b) Simpson diversity index:(1)
- c) Shannon diversity index: (2)
Response: We agree with your insightful points, We have added the calculation formula.
Calculate Simpson’s diversity and Shannon’s diversity index. It is defined as: D and
; (1)
; (2)
(Eqs. (1) and (2): denote the relative abundance of species inside the ith soil
- Yield measurements for Sapindus mukorossi were made during harvest at fruit maturity on 5 November 2021. The fruit was all ripe when the plants were picked. For fruit yield for each plot of Sapindus mukorossi involved picking all the ripe fruits from each whole tree, weighing the fruits using an electronic scale after picking, and then summing the fruit yield per tree for each plot.
Response: Thanks for your suggestion. We have revised this part.
Yield measurement: Yield measurements for Sapindus mukorossi were conducted on 5 November 2021, at the point of fruit maturity during harvest. All fruits were ripe at the time of harvesting the plants. To determine the fruit yield for each plot, all ripe fruits were collected from each entire tree, weighed using an electronic scale post-harvest, and subsequently, the yields per tree were summed for each respective plot.
- 3) On L87, delete space so change to: [11,14]
Response: Thank you. We have revised this part.
- Delete blank row on L195.
Response: Thank you. We have revised this part.
- On L201, delete space so change to: [2,28]
Response: Thank you for your suggestion, We have revised this part.
- On L342 this should be: Figure 3. Yield and…
Response: Thank you. We have revised this part.
- For ALL Tables, increase font size to Palatino Linotype size 10
Response: Thank you. We have revised Palatino Linotype size 10.
- Please check the English of all writing that was recently added to the manuscript. For example, the sentence added on L593-595 reads:
Response: Thank you. We have revised this part.
Meanwhile the research holds broader implications for sustainable agriculture and soil health management.
9. In the References, please put journal names in italics so for example Forest Ecol. Manag. on L560. This edit was not done.
Response: Thank you. We have checked all the references and the journal names are italicized
10. Author contribution on L583-595 lack all the specific roles and responsibilities from the original template. Please add these back in and specify which co-authors did what.
Response: Thank you. We have revised this part.

Reviewer 2 Report
Comments and Suggestions for Authors
The authors of the MS have addressed all my questions. I have no further comments. I recommend that the MS be accepted for publication.
Author Response
Dear Editor-in-Chief:
We have revised our previously-submitted manuscript entitled “Fertilization Induced Soil Microbial Shifts Show Minor Effects on Sapindus mukorossi Yield” (Manuscript ID: 3400875) by carefully following the comments by the two reviewers and the editor. Now, we resubmit the new manuscript based on the suggestions.
In this revised version we have responded to each of the issues raised by the reviewers to improve both the scope and expression of the work. The major changes in the current revision include:
(1)We have added calculation formulas and reorganized the sentence structure of the article as suggested by Reviewer 1. We have also made revisions to the Materials and Methods section as suggested by Reviewer 1, including modifying the entire text such as blank lines and fonts, removing unnecessary text, and refining the content to make it clearer.
(2)We have revised the abstract and discussion sections according to Reviewer # 2's questions: 'Some chapter abstracts should be moved to chapter introductions, and some chapters should be moved to chapter discussions'. Simplified abstract text and refined article content
The detailed responses to specific comments are listed below.
Yours Sincerely,
Juntao Liu
Ph.D.

Reviewer 3 Report
Comments and Suggestions for Authors
Some sections can be additionally enhanced to improve the integration of broader scientific contexts.
1. Incorporate the following key references to enhance the study's depth and contextual relevance:
DOI: 10.1016/j.heliyon.2021.e08142: Discuss broader soil microbial impacts on crop productivity.
DOI: 10.1016/j.scitotenv.2020.141687: Highlight the environmental implications of NPK fertilization strategies.
DOI: 10.9734/jabb/2024/v27i71104: Review recent advances in nutrient management for sustainable yield optimization.
DOI: 10.2741/4903: Explore microbial interactions with soil nutrient dynamics.
DOI: 10.1111/gcbb.12773: Investigate bioenergy crop management strategies under different nutrient regimes.
DOI: 10.1016/j.jbiotec.2022.04.002: Discuss advancements in biotechnological approaches for soil microbial enhancement.
DOI: 10.1016/J.APSOIL.2021.103970: Analyze soil acidification impacts under varying fertilization.
Except for these required references to be added, Authors can make additional effort and find more references to make the Introduction and Discussion section better.
2. Highlight the practical implications of the most effective NPK combination (N2P2K2).
3. Provide more nuanced discussions about microbial taxa showing differential responses to pH and nutrient changes. Use the listed references.
4. Ensure all statistical findings are explained in practical terms for readers less familiar with technical metrics.
5. Minor grammatical errors and typos (e.g., repeated phrases like "(4) (4)" on page 2). A professional edit can enhance readability.
6. Align figure captions with text descriptions for consistency.
Author Response
Dear Editor-in-Chief:
We have revised our previously-submitted manuscript entitled “Fertilization Induced Soil Microbial Shifts Show Minor Effects on Sapindus mukorossi Yield” (Manuscript ID: 3400875) by carefully following the comments by the two reviewers and the editor. Now, we resubmit the new manuscript based on the suggestions.
In this revised version we have responded to each of the issues raised by the reviewers to improve both the scope and expression of the work. The major changes in the current revision include:
(1)We have added calculation formulas and reorganized the sentence structure of the article as suggested by Reviewer 1. We have also made revisions to the Materials and Methods section as suggested by Reviewer 1, including modifying the entire text such as blank lines and fonts, removing unnecessary text, and refining the content to make it clearer.
(2)We have revised the abstract and discussion sections according to Reviewer # 2's questions: 'Some chapter abstracts should be moved to chapter introductions, and some chapters should be moved to chapter discussions'. Simplified abstract text and refined article content
The detailed responses to specific comments are listed below.
Yours Sincerely,
Juntao Liu
Ph.D.
Reviewer #3:
- 1. Incorporate the following key references to enhance the study's depth and contextual relevance:
DOI: 10.1016/j.heliyon.2021.e08142: Discuss broader soil microbial impacts on crop productivity.
DOI: 10.1016/j.scitotenv.2020.141687: Highlight the environmental implications of NPK fertilization strategies.
DOI: 10.9734/jabb/2024/v27i71104: Review recent advances in nutrient management for sustainable yield optimization.
DOI: 10.2741/4903: Explore microbial interactions with soil nutrient dynamics.
DOI: 10.1111/gcbb.12773: Investigate bioenergy crop management strategies under different nutrient regimes.
DOI: 10.1016/j.jbiotec.2022.04.002: Discuss advancements in biotechnological approaches for soil microbial enhancement.
DOI: 10.1016/J.APSOIL.2021.103970: Analyze soil acidification impacts under varying fertilization.
Response: Thank you for your valuable suggestions, which were very helpful in enhancing the depth and breadth of our study. We have already modified this part of the content.
- Highlight the practical implications of the most effective NPK combination (N2P2K2).
Response: Thank you for your careful review and suggestions. We have already modified this part of the content.
- Provide more nuanced discussions about microbial taxa showing differential responses to pH and nutrient changes. Use the listed references.
Response: Thank you for your review. We have already discussed it in the content of our article.
The study revealed that augmenting nitrogen application on the basis of P2K2 and phosphorus application on the basis of N2K2 notably decreased soil pH, likely due to the urea treatment exacerbatingIt was found that increasing N application on the base of P2K2 and P application on the base of N2K2 significantly reduced soil pH, probably because the urea treatment contributed more to soil acidification [30]. During the oxi-dation process of NH4+, H+is released back into the soil, exchanging cations that are absorbed by plants, thereby accelerating soil acidification [30-32].
- Minor grammatical errors and typos (e.g., repeated phrases like "(4) (4)" on page 2). A professional edit can enhance readability.
Response: Thank you for your your suggestions of what we have added to the manuscript. This is the introduction to the content of this article, so it needs to be cited again [4]
- Align figure captions with text descriptions for consistency.
Response: Thank you. We have revised this part.
